# The genomic and transcriptional landscape of primary central nervous system lymphoma

Josefine Radke [1,2,3,68✉], Naveed Ishaque [4,68✉], Randi Koll[1,2], Zuguang Gu[5], Elisa Schumann[1,2], Lina Sieverling [6,7], Sebastian Uhrig[5,8], Daniel Hübschmann [5,8,9,10,11], Umut H. Toprak[12], Cristina López[13,14], Xavier Pastor Hostench [5], Simone Borgoni[15], Dilafruz Juraeva [5], Fabienne Pritsch[1,2], Nagarajan Paramasivam[5], Gnana Prakash Balasubramanian[16,17], Matthias Schlesner [18,19], Shashwat Sahay[3,4], Marc Weniger[20,21], Debora Pehl[1], Helena Radbruch [1], Anja Osterloh[1], Agnieszka Korfel[22], Martin Misch[23], Julia Onken[23], Katharina Faust[23], Peter Vajkoczy[23], Dag Moskopp [24], Yawen Wang[24], Andreas Jödicke[25], Lorenz Trümper[26], Ioannis Anagnostopoulos[27], Dido Lenze[27], Ralf Küppers [20,21], Michael Hummel[27], Clemens A. Schmitt [28,29], Otmar D. Wiestler[30], Stephan Wolf[31], Andreas Unterberg[32], Roland Eils [4], Christel Herold-Mende[32], Benedikt Brors [5], ICGC MMML-Seq Consortium*, Reiner Siebert[13,69], Stefan Wiemann [11,15,69] & Frank L. Heppner [1,2,33,34,69]

Primary lymphomas of the central nervous system (PCNSL) are mainly diffuse large B-cell lymphomas (DLBCLs) confined to the central nervous system (CNS). Molecular drivers of PCNSL have not been fully elucidated. Here, we profile and compare the whole-genome and transcriptome landscape of 51 CNS lymphomas (CNSL) to 39 follicular lymphoma and 36 DLBCL cases outside the CNS. We find recurrent mutations in JAK-STAT, NFkB, and B-cell receptor signaling pathways, including hallmark mutations in *MYD88* L265P (67%) and *CD79B* (63%), and *CDKN2A* deletions (83%). PCNSLs exhibit significantly more focal deletions of HLA-D (6p21) locus as a potential mechanism of immune evasion. Mutational signatures correlating with DNA replication and mitosis are significantly enriched in PCNSL. *TERT* gene expression is significantly higher in PCNSL compared to activated B-cell (ABC)-DLBCL. Transcriptome analysis clearly distinguishes PCNSL and systemic DLBCL into distinct molecular subtypes. Epstein-Barr virus (EBV)+ CNSL cases lack recurrent mutational hot-spots apart from IG and *HLA-DRB* loci. We show that PCNSL can be clearly distinguished from DLBCL, having distinct expression profiles, *IG* expression and translocation patterns, as well as specific combinations of genetic alterations.

A full list of author affiliations appears at the end of the paper.

Central nervous system (CNS) lymphomas are predominantly aggressive neoplasms involving brain, meninges, spinal cord, and eyes[1,2]. Two clinical subtypes of CNSL can be distinguished: primary central nervous system lymphoma (PCNSL), which is confined to the CNS; and secondary central nervous system lymphoma (SCNSL) presenting initially with systemic, non-CNS or synchronous systemic and CNS involvement. The term SCNSL comprises all systemic lymphomas that spread to the CNS and its presentation, tropism, outcome and therapeutic options differ from PCNSL[3,4]. Typically, SCNSL are classified as diffuse large B-cell lymphoma (DLBCL), while other types such as follicular lymphoma (FL), T-cell lymphoma or Hodgkin lymphoma are extremely rare[5,6].

PCNSL incidence is increased in immunocompromised patients, in which the tumor cells are typically Epstein-Barr virus (EBV)-positive[1,7,8]. In contrast, PCNSL in immunocompetent patients is typically EBV-negative. The mechanisms leading to the observed exclusive topographical restriction of PCNSL to the CNS are not fully elucidated[9]. PCNSL is classified as DLBCL in the vast majority of cases (approx. 90%) which immunohistochemically most often show a non-germinal center B-cell-like (non-GCB) immunophenotype[1,10,11] according to the Hans classification[12]. The tumor cells express pan B-cell markers (CD19, CD20, and CD79a), the germinal center (GC)-associated molecule BCL6[13], and the post-GC-associated marker MUM1/IRF4[14]. By gene expression profiling, the tumor cells are most closely related to late germinal center (exit) B-cells[15]. Pathomechanistic genomic alterations involving Toll-like- and B-cell receptor (TLR, BCR) signaling pathways are identified in previous studies revealing a very high frequency of somatic nonsynonymous mutations in genes such as *MYD88, CARD11*, and *CD79B*[16–20]. Additionally, often homozygous HLA class II[21,22] and *CDKN2A* loss, recurrent *BCL6* translocations[23,24] and structural variants at chromosome band 9p24.1 (affecting *CD274/PD-L1 and PDCD1LG2/PD-L2*)[25] as well as *TBL1XR1* variants[26] are repeatedly described in PCNSL[27,28]. These mutational patterns suggest PCNSL to be genetically similar to recently described "MCD", "C5" or "MYD88-like" subtypes for which a derivation from long-lived memory B-cells is proposed[29–35].

The outcome of PCNSL, even in immunocompetent hosts, is poor compared to most primary systemic DLBCL[36], though probably not worse than that of DLBCL of the MCD/C5 group in general[30]. High-dose methotrexate (MTX) remains the commonly administered therapy but the use of rituximab (monoclonal anti-CD20 antibody) is shown to be effective[37,38]. However, reports on rituximab efficiency in PCNSL are conflicting[39–42]. Genomic studies suggest that lymphoma cell proliferation and survival are driven at least in part, by deregulated TLR, BCR, JAK-STAT, and NFκB signaling pathways inducing constitutive NFκB activation[43–45]. Therefore, inhibitors up- and downstream of NFκB such as ibrutinib, known to inhibit Bruton's tyrosine kinase (BTK) as critical mediator of BCR signaling, and lenalidomide which is shown to have indirect effects on tumor immunity are applied and seem to be effective therapeutic alternatives in PCNSL[46–51]. PD-L1/2 blockade is discussed as another therapeutic option[52].

Despite all progress in the molecular characterization of PCNSL in the last decades, our understanding of the genetic and transcriptional alterations of PCNSL is by far not comprehensive. The few previous next generation sequencing (NGS) studies of PCNSL are limited to target enrichment only of exons, or whole-genome analysis of very few samples[44,53–58]. Therefore, pathogenic mechanisms other than coding variants, such as non-coding and regulatory changes, structural variants or mutational mechanisms related to the genome-wide distribution of somatic hypermutation (SHM) are not fully elucidated in PCNSL.

Unbiased omics profiling, such as whole-genome sequencing (WGS) studies integrated with transcriptome sequencing, are currently the methods of choice to illuminate the role of non-coding mutations[59,60]. In addition, these approaches can unravel various molecular mechanisms deregulating driver genes in PCNSL, which are necessary for diagnosis, risk stratification, and treatment in the era of precision and targeted therapies.

In this work, we perform whole-genome and transcriptome sequencing in 51 B-cell lymphomas presenting in the CNS, including 42 PCNSL samples from immunocompetent patients, to comprehensively describe the mutational and transcriptional landscape of PCNSL.

## Results

**Study cohort.** We enrolled CNSL samples from 51 adults diagnosed with PCNSL or SCNSL. According to the site of manifestation, the following subgroups were defined: PCNSL within the brain parenchyma (PCNSL; $n = 39$), PCNSL with meningeal manifestation (PCNSL-M; $n = 3$), SCNSL within the brain parenchyma (SCNSL; $n = 3$), SCNSL with meningeal manifestation (SCNSL-M; $n = 3$) and EBV-positive lymphomas (EBV+; $n = 3$). Median age was 69, mean age was 66.5 years at diagnosis (range 40–82 years). The female:male ratio was 1.3:1. Follow up data were available for 44 patients. The follow up time ranged from 1 to 104 months with a median survival of 15.0 months (Supplementary Fig. 1a). The detailed study cohort information and subgroup-specific demographics are given in Fig. 1a, b and Supplementary Data 1. Patient samples were histologically and immunohistochemically classified according to the WHO criteria[2,11,61,62], and further stratified according to the Hans classification[12] into non-GCB ($n = 37$) and GCB subgroup ($n = 5$, Fig. 1c, Supplementary Data 1). For nine samples, the tissue was not sufficient for non-GCB/GCB characterization. Furthermore, we integrated data from the ICGC MMML-Seq cohort (www.icgc.org) for comparison of WGS and transcriptome data from systemic DLBCL, FL, naïve B-cells, and GC B-cells[34,59,60].

**Mutational landscape of central nervous system lymphoma (CNSL).** WGS data of 38 CNSL (30 PCNSL, 1 PCNSL-M, 2 SCNSL, 3 SCNSL-M, and 2 EBV+ samples, Fig. 1b) was obtained with a median coverage of 77 (range 31–100) for tumors and 45 (range 27–85) for matched germline controls. We identified a median of 18584 (range: 1987–48280; median of the 30 PCNSL: 20263 (range: 9185–48280)) total SNVs, of which a median of 5759 (range: 686–16731; PCNSL: 6274 (range: 2850–16731)) were intronic, a median of 10218 (range: 983–24033; PCNSL: 10790 (range: 5063–24033) were intergenic, and a median of 194 (range: 47–436), PCNSL: 200 (range: 100–436) were nonsynonymous exonic variants (1% of all SNVs). Furthermore, we identified a median of 2406 (range: 711–9430; PCNSL: 2485 (range: 941–9430)) indels per CNSL sample, of which the majority was intergenic (1333 (range: 403–5218), PCNSL: 1375 (range: 517–5218). The median number of variants (SNVs and indels) in non-coding RNA genes was 2744 (range: 551–6913), PCNSL: 2901 (range: 1220–6913). Selected variants were verified using Sanger sequencing (see "Methods" section).

The CNSL cohort presented a median of 152 (PCNSL: 147) SVs (range: 24–517 (PCNSL: 47–517), inversions: 21 (PCNSL: 20), deletions: 76 (PCNSL: 81), duplications: 20 (PCNSL: 19), translocations: 14 (PCNSL: 16)). We also investigated chromosome level CNVs (based on 30% or more of a chromosome being amplified or deleted) and found a median of 8 CNVs (median 1 cnLOH (PCNSL: 2), median 4 gains (PCNSL: 4), and median 2 losses (PCNSL: 3)). The detailed mutational statistics (CNVs,

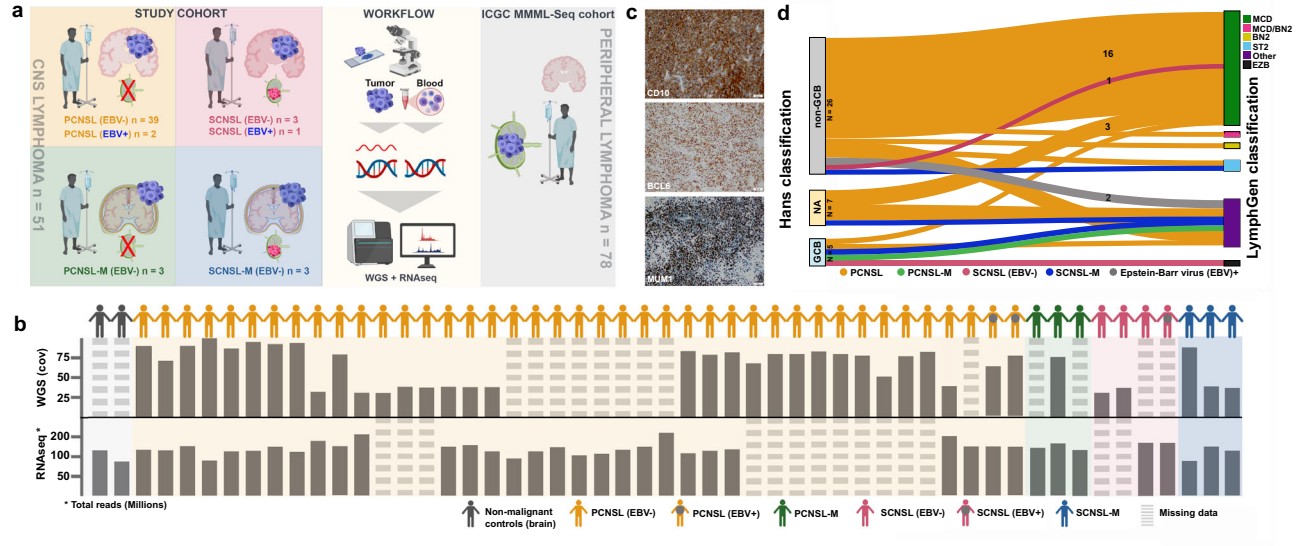

**Fig. 1 Study design and multi-omic analysis of the CNSL cohort.** Panel **a** demonstrates the study design, study cohort along with sample size, and sequencing approach. The central nervous system lymphoma (CNSL) cohort consists of 51 primary and secondary lymphoma (PCNSL and SCNSL) patients. Whole-genome sequencing was performed using tumor tissue and matched peripheral blood samples. For RNA sequencing, we additionally analyzed normal controls (non-malignant brain tissue (frontal lobe)) and included data from peripheral lymphomas without CNS manifestation for validation (ICGC MMML-seq cohort). The lower panel **b** lists all CNS lymphoma derived from PCNSL or SCNSL patients as well as controls. The depth of coverage (cov) for whole-genome sequencing and total number of reads in RNA sequencing are given for each tumor sample. Whole-genome sequencing data were obtained from $n = 38$ PCNSL/SCNSL patients. RNA sequencing data was generated from $n = 37$ PCNSL/SCNSL patients and $n = 2$ normal controls. In 24 PCNSL/SCNSL cases, we obtained patient-matched whole-genome and RNA sequencing data. Striped bars indicate data sets missing for individual samples. The CNSL specimen were examined according to the Hans classification (CD10, BCL6, and MUM1 (**c**)). Additionally, we classified all $n = 38$ whole-genome sequencing CNSL samples according to the genetic DLBCL subtypes using the LymphGen algorithm as described by Wright et al., 2020 (**d**). The results are displayed in a Sankey plot. Images in panels **a** and **b** were partially created with BioRender.com.

indels, SNVs, and SVs) of the CNSL, DLBCL, and FL samples are displayed in Supplementary Data 2.

**PCNSL represents MCD genetic subtype of DLBCLs.** Recent exome studies described the existence of different genetic subtypes of DLBCL, which show activation of distinct signaling pathways and different clinical outcomes[25,30,31]. We used the LymphGen algorithm described by Wright et al.[31] to classify our samples according to these genetic subtypes based on the obtained WGS data. The results of the CNSL cohort are displayed in Fig. 1d. In line with previous results[17,20,30,31], the majority of PCNSL samples were classified as MCD (based on the co-occurrence of *MYD88* L265P and *CD79B* mutations, 67%, 20/30). One sample was each assigned to BN2 (*BCL6* fusions and *NOTCH2* mutations, 3%, 1/30) and ST2 (*SGK1* and *TET2* mutated, 3%, 1/30), seven samples were non-subtyped cases ("Other", 23%, 7/30), and one sample was equally assigned to both groups BN2/MCD (3%, 1/30; Supplementary Data 1).

PCNSL samples classified as "Other" exhibited different CNV profiles affecting chromosome arms 1q, 2p, 2q, 3q, 4p, and 11p, as well as significantly more deletions of *CREBBP* compared to PCNSL samples classified as MCD by the LymphGen algorithm (Supplementary Fig. 2a). *CREBBP* gene inactivation is considered an early event in FLs and a subset of systemic DLBCL, mostly of GCB origin[63–67]. *CREBBP* inactivation is also described as a hallmark of the EZB class, but LymphGen's classification model is restricted to *CREBBP* point mutations and not focal deletions. The finding of a significantly increase number of *CREBBP* alterations ($p = 0.046$, Mann–Whitney U test) in PCNSLs classified as "Other" compared to MCD might, thus, imply a small subset of PCNSL to more resemble GCB-like DLBCL or, alternatively, the existence of a group of occult systemic

GCB-lymphomas with first clinical presentation in the CNS. Additionally, PCNSL-Other demonstrated significantly fewer mutations in *GRHPR, ETV6*, and *PIM1* (Supplementary Fig. 2b, c).

**Driver mutations in CNSL.** We first identified the genes recurrently mutated in CNSL (Fig. 2a) and used Metascape[68] for further pathway and process enrichment analysis. The top three level enriched terms were 'Regulation of hemopoiesis', 'Chromatin organization involved in negative regulation of transcription', and 'Cytokine signaling in immune system' ((hypergeometric test, FDR $8.91 \times 10^{-9}$, $1.04 \times 10^{-4}$, $1.17 \times 10^{-4}$, respectively; Fig. 2b). The enrichment analysis in TRRUST revealed 'Regulated by: STAT3' as the most significant term (hypergeometric test, FDR $3.98 \times 10^{-7}$; Fig. 2c). *STAT3* has been associated with intracranial spreading and poor survival in PCNSL[69,70], and reports of STAT3 inhibition via small molecules achieve complete tumor regression in vivo for lymphoma cell lines[71]. As *STAT3* is not highly mutated or hit by SVs or CNVs, its activation seems—in line with previous reports—induced by extrinsic factors such as infiltrating macrophages/microglial cells[72], or intrinsic factors such as activation downstream of MYD88[73].

Next, we used IntOGen and MutSigCV to discover putative driver mutations in the PCNSL WGS sub-cohort (Fig. 2d and Supplementary Data 3). We identified a total of 50 mutated driver genes, of which only 21 were previously known drivers. Many of the predicted drivers were associated with MCD enriched genes, including *MYD88* (67%), *CD79B* (63%), *OSBPL10* (83%), *HLA-A/B/C* (40%/63%/53%), *PRDM1* (40%), *TOX* (50%), *TBL1XR1* (40%), *CD58* (37%), *PIM1* (70%), *ETV6* (50%), *BCL11A* (30%), *CDKN2A* (83%), *GRHPR* (60%), *FOXC1* (20%), *and DAZAP1* (20%). These driver genes were significantly enriched for genes containing the *BCL6* binding motif

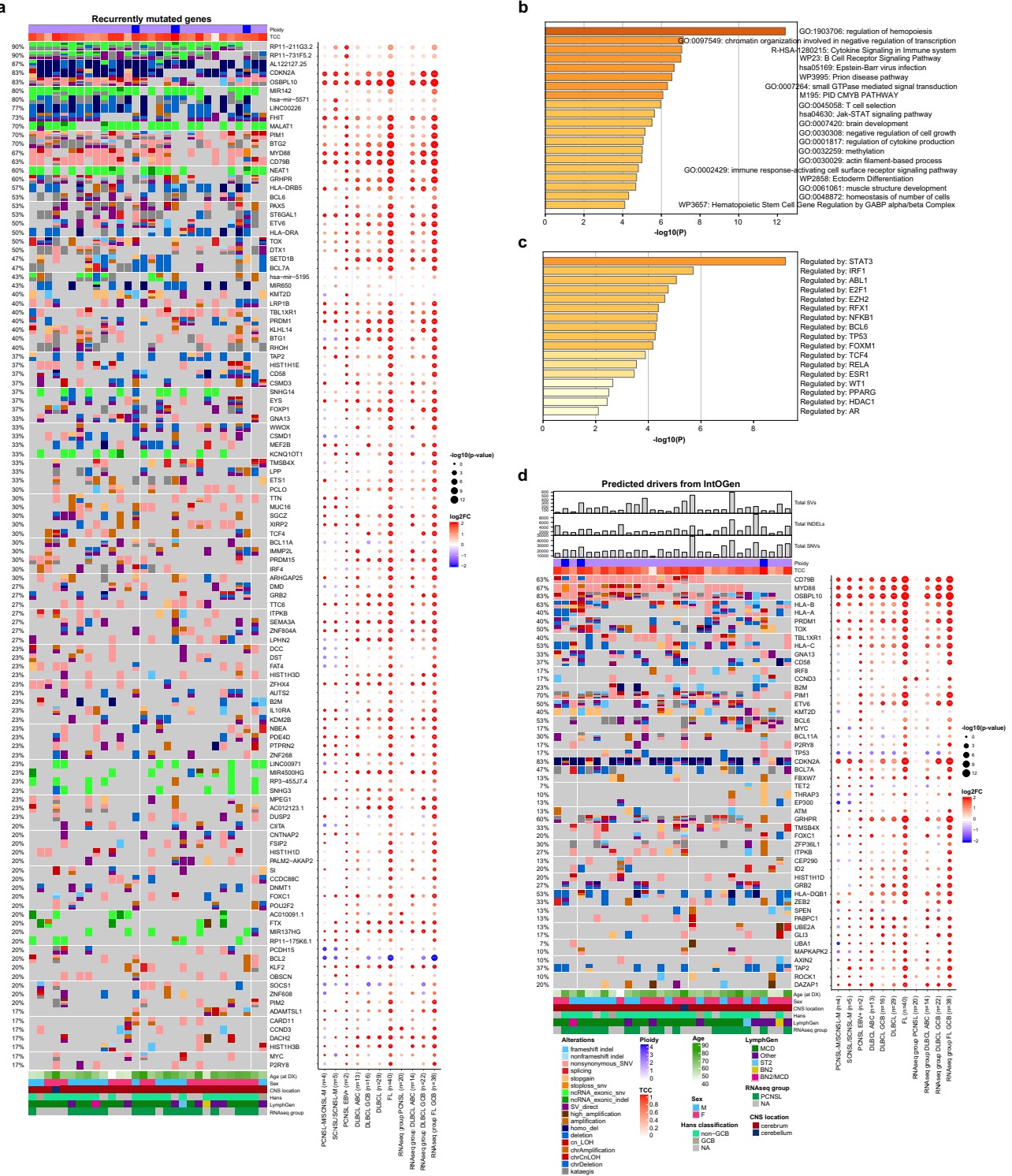

**Fig. 2 Recurrent coding mutations and genomic drivers in PCNSL.** Oncoprints of recurrently mutated genes, excluding IG and chromosome Y genes (**a**). Mutated genes are listed from top to bottom depending on their alteration frequency. The corresponding dot plot reflects the log2 fold change and significance of alteration frequencies in the other subcohorts and RNAseq subgroups compared to PCNSL. The recurrently mutated genes (listed in the oncoprint in a) were analyzed by Metascape[68] to identify pathway and process enrichment (**b**) and transcriptional regulatory networks (TRRUST) (**c**). Metascape adopts the hypergeometric test and employs the Benjamini-Hochberg correction for multiple testing. Oncoprints of driver genes in PCNSL (**d**). The top panel of the oncoprint shows the total numbers of structural variants (SVs), small insertions/deletions (INDELs), single nucleotide variants (SNVs), estimated ploidy, and genomic tumor cell content (TCC). Mutated genes are ranked by IntOGen. The corresponding dot plot reflects the log2 fold change and significance of alteration frequencies in the other subcohorts and RNAseq subgroups compared to PCNSL. In panels (**a**) and (**d**) the size of the dots demonstrate the significance according to a two-tailed Fisher's exact test not corrected for multiple testing.

(TRANSFAC and JASPAR PWMs, Enrichr enrichment test, adjusted $p = 0.03193$).

*OSBPL10* was previously reported as a target of aSHM in PCNSL[53]. Consistent with observations in DLBCL[74], most of the identified mutations in PCNSL were confined to the exon 1 coding region (Supplementary Fig. 3).

Concerning *MYD88*, we only detected the classical pathogenic hotspot L265P mutation, which was validated by Sanger sequencing in all PCNSL samples investigated (100%, $n = 26$, Supplementary Data 4 and 5). Notably, *MYD88* mutation rates in the extension FFPE cohort were 78.9% in PCNSL and 55.6% in SCNSL. None of the five EBV-positive cases investigated harbored oncogenic *MYD88* L265P mutations, which is in line with previous findings[25,75]. Mutations in *TBL1XR1* also modulating TLR/MYD88 signaling[25] were identified in 40% of PCNSL (Fig. 2a, b). We investigated mutual exclusivity and co-occurrence patterns for *MYD88* among the driver genes that affect at least five patients using Fisher and CoMET test. We observed mutual exclusivity between alterations in *MYD88* and the NOTCH signaling inhibitor *SPEN*[30] (Fisher test, $p = 0.0009$, FDR = 0.033). In line with previous reports on ABC-DLBCLs[76,77], we found coexisting alterations in *MYD88* and *CD79B*. Nevertheless, this co-occurrence was not significant (Fisher test, $p = 0.16$, FDR = 1.0). However, *MYD88* was most significantly co-occurring with *TBL1XR1* (Fisher test, $p = 0.04$), both activating the NFκB signaling pathway[26]. Although this was not significant after correction for multiple testing (FDR = 1, Supplementary Data 6).

Compared to the MCD driver genes identified in the series presented by Wright et al.[31], our PCNSL series exhibited a higher proportion of samples with mutations in *PABPC1* (10% vs 0%), *P2RY8* (13% vs 1.2%), *ITPKB* (23% vs 2.5%), *GNA13* (20% vs 5.1%), and *B2M* (13.3% vs 2.8%). Furthermore, predicted driver genes in our PCNSL series included genes enriched in all other LymphGen classes: BN2 (*CCND3, BCL6, HIST1H1D, SPEN, PABPC1,* and *UBE2A*), EZB (*GNA13, IRF8, BCL7A, KM2TD,* and *EP300*), ST2 (*P2RY8, TET2, ZFP36L1,* and *ITPKB*) and A53 (*B2M* and *TP53*).

While the majority of identified drivers were reported by Wright et al., a number were not, including *FBXW7, ATM, TMSB4X,* THRAP3, *ID2, GRB2, ZEB2, GLI3, UBA1, MAPKAPK2, AXIN2, TAP2, ROCK1, CEP290,* and *HLA-DQB1*. These were previously recognized as general DLBCL drivers by Reddy et al.[78] and/or Chapuy et al.[29]. *ZEB2* was additionally identified as a genetic alteration associated either with the ABC subgroup[78] or the DLBCL C1 cluster[29] (Supplementary Data 3).

A remarkable finding was the identification of *MYC* mutations in 17% of PCNSL in the absence of *MYC* translocations. *MYC* alteration does not belong to the defining feature of the LymphGen algorithm nor has it been described as a driver in DLBCL by Chapuy et al.[29], though its functional relevance as oncogene in DLBCL has been shown by Reddy et al.[78]. Mutation of *MYC* in lymphomas is frequently linked to IGH translocations, which nevertheless are rare in the PCNSL as shown in the present as well as previous studies[23,79]. Whereas previous studies showing a high frequency of *MYC* mutations in PCNSL focused on the region underlying SHM in PCNSL[80], we here show that these mutations scatter across the gene (Supplementary Fig. 4). The function of the changes remains elusive but it is intriguing to speculate that at least part of them might contribute to the "double expression" of BCL2 and MYC in the absence of *MYC* translocation in PCNSL which has been associated with unfavorable outcome in systemic DLBCL[81].

**Recurrent somatic alterations in non-protein-coding genes**. The landscape of mutations affecting ncRNA in PCNSL was comparable to ABC-DLBCL, apart from significantly more mutations in *AL122127.1* and *AL122127.4* (Fig. 3a), situated in the *IGH* locus, and in *RP11-211G3.2*, situated in the first intron of *BCL6*. While the implications of these mutations are unclear, it is possible that these mutations are accumulated as part of the SHM/aSHM process affecting *IGH* and *BCL6*. Additionally, we identified recurrent aberrations in the aSHM target *MIR142* (80%; Fig. 3a, b) as well as *MALAT1* (70%) and *NEAT1* (60%), both located 53 kb apart on 11q13.1. The mechanistic roles of many ncRNAs are poorly understood because their exact function is difficult to assess. However, the lncRNAs *NEAT1* (nuclear enriched abundant transcript 1) and *MALAT1* (metastasis-associated lung adenocarcinoma transcript 1) are well known to play essential roles in the development and progression of various cancers by influencing gene expression by alternative splicing and epigenetic modification of regulatory elements[82–84]. Both, *MALAT1* and *NEAT1*, which have not been linked to PCNSL before are known to be mutated and highly expressed in DLBCL[34] and predict poor prognosis[85,86]. Further aberrations in lncRNAs affected *KCNQ1OT1* (33%) and *SNHG3* (23%), both reported to have oncogenic functions in multiple cancers[87,88] as well as *SNHG14* (37%), promoting immune evasion in DLBCL[89].

**Kataegis shapes the mutational repertoire of PCNSL**. Kataegis is a pattern of mutational hotspots that has been associated with a number of cancers[90], and is a frequent consequence of AID activity in lymphomas[91]. Many of the recurrently mutated genes in PCNSL were dominated by alterations that are located in these highly mutated hotspots[11], of which several have previously been described as targets of aSHM, such as *OSBPL10, PIM1, BTG2,* and *PAX5* (Fig. 3b)[25,53,80]. Of the 50 identified protein-coding driver genes and the top 50 mutated ncRNA in PCNSL, 15 and 21 were targeted by kataegis, respectively (Figs. 2a, b, 3a, b, additional supplement [https://doi.org/10.5281/zenodo.6054242][92]). Consistent with previous reports[93], expression of miRNA, lncRNA, antisense RNA, and protein coding genes with kataegis loci were expressed significantly higher than those without (Wilcoxon rank sum test, $p < 0.05$; Fig. 3c). This implicates that either aSHM preferentially targets highly expressed genes, or that aSHM may cause hyperactivation of these genes. Interestingly, the largest difference in RNA expression was observed in miRNA genes, again highlighting the importance of the non-coding alterations in PCNSL. This observation was consistent for subgroups, including systemic DLBCL (Supplementary Fig. 5).

Physiologically, SHM is the process of introducing mutations in the antibody genes to alter the antigen-binding site, increasing the immunoglobulin (IG) diversity[94]. Kataegis events were at *IGH* (100%), *IGL* (100%) and *IGK* (70%) loci but were also found outside IG loci, targeting *BTG2* (63%), *GRHPR* (50%), *PIM1* (43%), *DTX1* (40%), *OSBPL10* (37%), *ZNF860* (37%), *BCL6* (33%), *RHOH* (33%), *CXCR4* (30%), *BACH2* (27%), and *PAX5* (27%; Fig. 3b). The recurrently targeted genes in PCNSL mostly overlapped with those targeted in ABC-DLBCLs. However, samples with mutational hotspots in *BTG2, GRHPR, OSBPL10* and *ZNF860* were significantly more frequent in PCNSL (in 18, 15, 11 and 11 of 30 samples, respectively) compared to ABC-like DLBCL (in 2, 0, 0 and 0 of 13 samples, $p = 0.009, 0.001, 0.019$ and 0.019, Fisher's exact test, respectively). Taking all non-IG genes that overlapped a mutational hotspot in at least one PCNSL sample (242 genes, Supplementary Data 7), we found the BCR signaling pathway to be most significantly enriched (Enrichr enrichment test, adjusted $p = 0.0046$). Taken together, kataegis and aSHM play a decisive role in shaping the mutational repertoire of PCNSL and are associated with functional pathways in PCNSL pathogenesis.

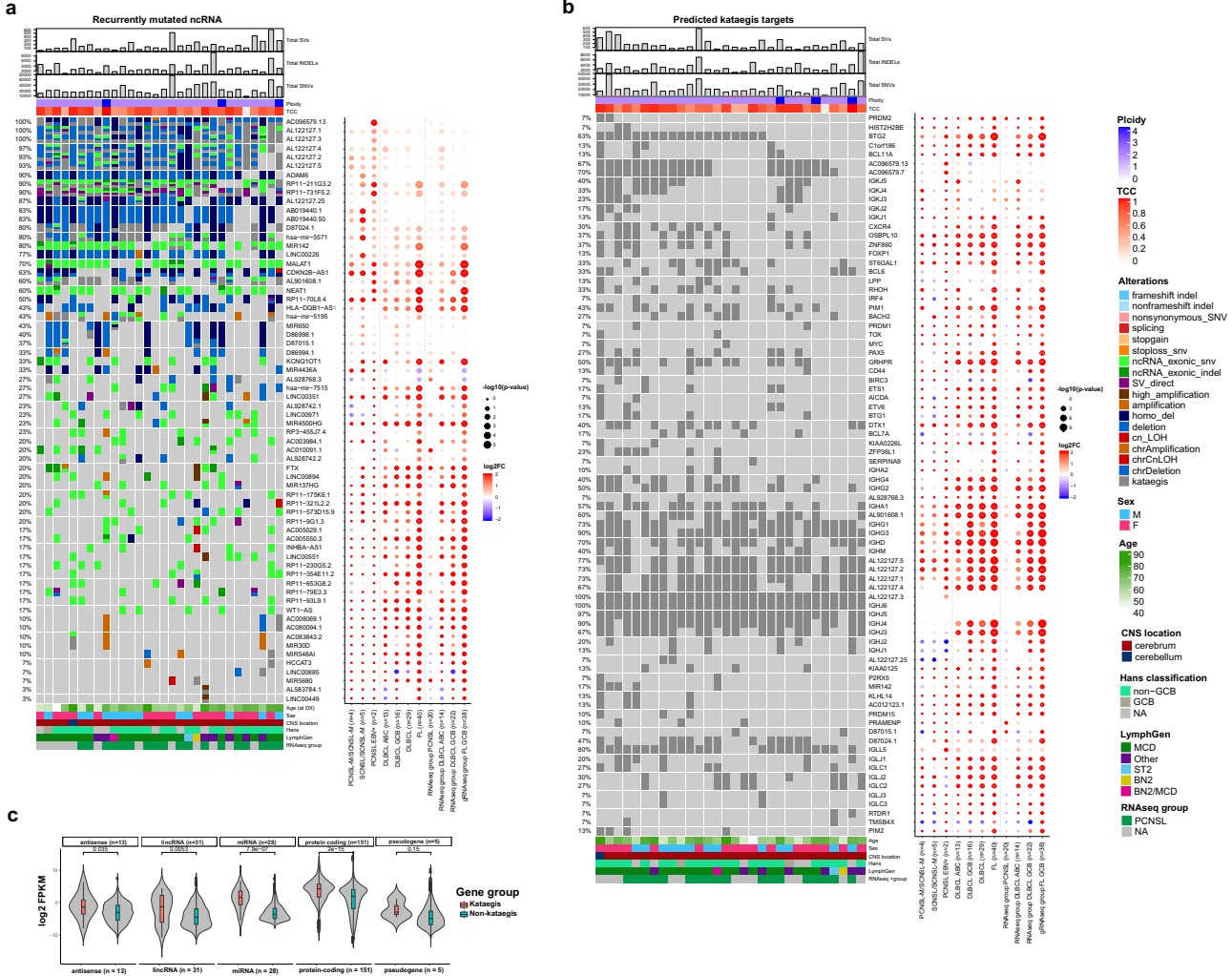

**Fig. 3 Recurrent non-coding RNA mutations and Kataegis events in PCNSL.** Oncoprint of recurrent non-protein-coding genes in PCNSL (**a**). Shown are the total numbers of structural variants (SVs), small insertions/deletions (INDELs), single nucleotide variants (SNVs), estimated ploidy, and tumor cell content (TCC) (top panel). Mutated genes are listed from top to bottom depending on their alteration frequency. Oncoprint of recurrent kataegis events in PCNSL (**b**). Mutated genes are listed from top to bottom depending on chromosome location. The corresponding dot plot reflects the log2 fold change and significance of alteration frequencies in the other subcohorts and RNAseq subgroups compared to PCNSL. In panels **a** and **b** the size of the dots demonstrate the significance according to a two-tailed Fisher's exact test not corrected for multiple testing. The violin plot **c** shows that the RNA expression of genes with kataegis loci compared to those without were significantly higher for antisense, long non-coding RNA, miRNA and protein coding genes (one-sided Wilcoxon rank sum test, $p < 0.05$). Box and whisker plots are embedded in the violin plots, showing median (center line), the upper and lower quartiles (the box), and the range of the data (the whiskers), excluding outliers.

While patterns of aSHM and kataegis were similar between CNSL and systemic DLBCL subtypes, we identified that EBV+ CNSL cases did not share many of the recurrent mutational hotspots apart from IGH and the *HLA-DRB* locus. (Fig. 3b, Supplementary Fig. 6a, b, Supplementary Data 5).

**Recurrent copy number alterations (CNAs).** Compared to systemic ABC-DLBCL and GCB-DLBCL of which the copy number profiles reflected previously published results[34,60,95], PCNSL demonstrated significantly more CN losses in 6p21 (*HLA-D* locus, Fig. 4a–c, Supplementary Data 8) as well as recurrent losses in 9p21 (*MTAP*, *CDKN2A/B*) and 19p13 (*CDKN2D*). The loss of the *HLA-D* locus that encode for MHC class II molecules lead to reduced immune surveillance and poor survival in DLBCL[96]. *CDKN2A* is an established tumor suppressor gene with roles in angiogenesis, cell death, invasiveness, and growth suppression[97–99]. Additionally,

we found deletions on chromosomes 1p13 and 3q13, affecting genes such as *CD58* and *CD80*, both candidates reported to lead to immune evasion[100]. Further CN losses were detected on chromosomes 8q12 (*TOX*), 12p13 (*ETV6*), and 15q21 (*B2M*) as well as 3p14, affecting the fragile site tumor suppressor gene, fragile histidine triad (*FHIT*). *TOX* deletions have been previously described by array-based imbalance profiling[101]. *TOX* is required for the development of various T-cell subsets and was described as putative tumor suppressor in MCD DLBCL[30]. *TOX* downregulation has been associated with poor prognosis in different cancers[102] and is a predictor for anti-PD1 response[103]. Significant CN gains in PCNSL mapped to 2q37 and 18q21 affecting *DIS3L2* and *MALT*. *DIS3L2* encodes for an exoribonuclease that is responsible for Perlman syndrome[104] and was recently described to promote HCC tumor progression by upregulating production of the oncogenic isoform of *RAC1*, *RAC1B*[105]. *MALT* is a regulator of NFκB signaling and potential therapeutic target in B-cell lymphoma[106].

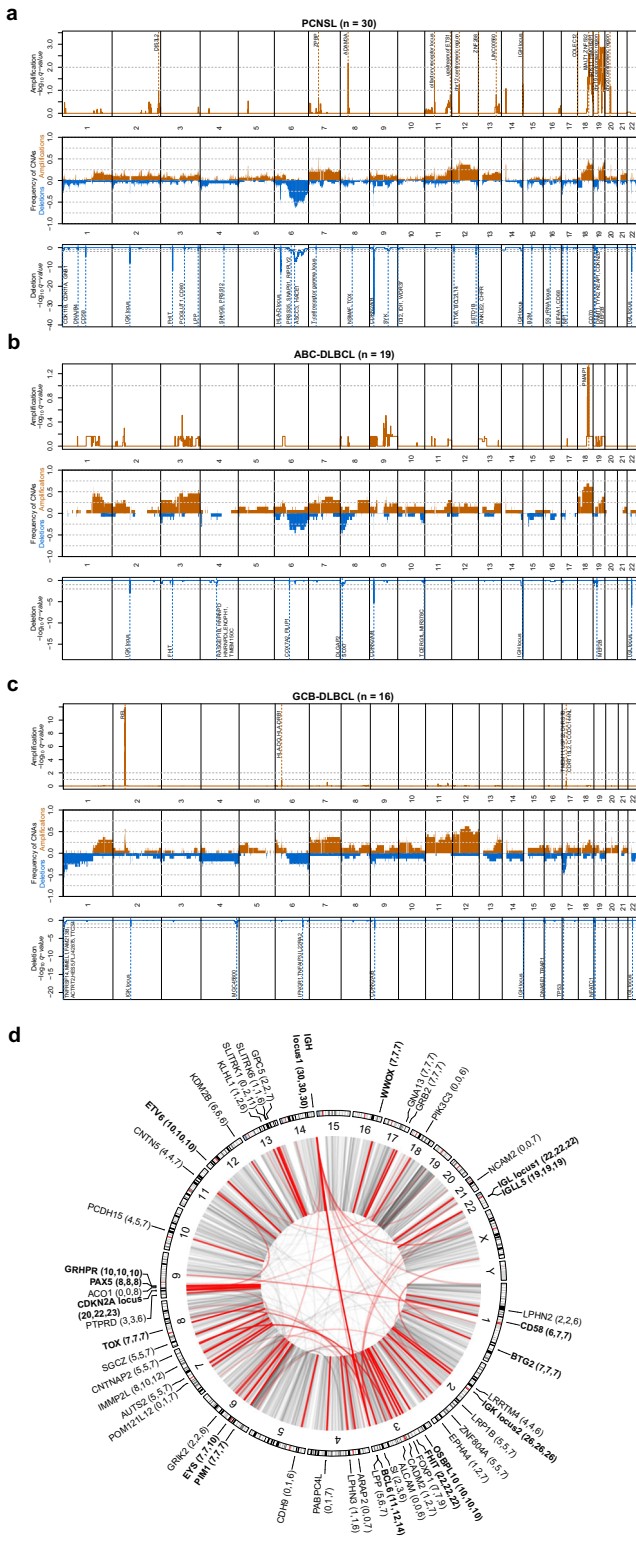

**Fig. 4 Genomic structural variation in PCNSL.** Recurrent somatic CNAs in PCNSL (**a**), ABC-DLBCL (**b**), and GCB-DLBCL (**c**). Relative prevalence of somatic copy number aberrations in tumor samples (middle panel), showing the presence of at least one copy number gain (orange bars), copy number loss (blue bars), as a proportion of analyzed samples. Significantly (q-value < 0.25) amplified and deleted regions and candidate genes are shown (upper and lower panel). For GCB-DLBCL, we added TP53 as this was detected in a significant broad deletion (Gistic2 p-value 0.0311), and the focal peak falls on the region including TP53. Circular visualization of genome rearrangements in PCNSL (**d**). The panels (from outside going inwards) represent recurrence per gene, chromosome ideograms, and chromosome numbering. Next to gene names, the number of SV breakpoints that lie direct on the gene, within 100 kbp of the genes, and closest to that gene are reported. Inter-chromosomal translocations are rendered with black (all) and red (highlighted if affected in 20% of samples) arcs in the center of the display. The significance of CNVs was calculated using the GISTIC 2.0 permutation test with Benjamini–Hochberg correction for multiple testing.

**Recurrent structural variations (SVs).** We defined SVs as genomic breakpoints, which can correspond the borders of amplifications and deletion, but also balanced translocations and inversions. PCNSL showed a median of 147 SVs (range: 24–517, Supplementary Data 2). IG gene rearrangements were found in all PCNSL, ABC-DLBCL, and GCB-DBCL cases and affected the *IGH* (100, 100, and 100%), *IGL* (73, 46, and 31%) and *IGK*

(87, 54, and 63%) loci. Furthermore, direct SVs affected *FHIT* (73, 23, and 38%), *CDKN2A* (67, 38, and 25%), *BCL6* (37, 21, and 19%), *OSBPL10* (33, 8, and 13%), *ETV6* (33%, 15%, 6%), *PAX5* (27, 0, and 13%), *PIM1* (23, 0, and 6%), *TOX* (23, 8, and 19%), *BTG2* (23, 8, and 0%), *WWOX* (23, 8, and 25%), as well as *CD58* (20, 8, and 19%; Fig. 4d, Supplementary Data 9). *WWOX* and *FHIT* represent common fragile site (CFS) and have been classified as tumor suppressor genes in DLBCL[107,108].

Recent studies have shown that translocation can act as enhancer hijacking even when the events is several hundred thousand base-pairs away from target genes[109]. To investigate this, we also annotated SV breakpoints to genes within 100 kbp and also to the closest genes. We found a number of genes involved in G protein-coupled receptor signaling (*ARAP2, LPHN2, LPHN3, EPHA4, ADGRL2,* and *GPC5*) consistent with observations in pan-cancer studies[110]. A number of other genes exhibited at least three times as many distal translocations (while still being the closest gene) than directly on the gene, including *PIK3C3, EPHA4, SI, ALCAM, NCAM2, CADM2, CDH9, PABPC4L, GRIK2, POM121L12, ACO1, KLHL1, SLITRK1,* and *SLITRK6*. Hyperactivation of PI3K signaling is one of the most common events in human cancers, and PIK3C3 has been shown to promote cell proliferation[111] and autophagy[112], and its inhibition has shown therapeutic benefit in bladder, hepatocellular (HCC), and colon cancer[113–115]. *EPHA4* has been described to promote cell proliferation and migration[116,117] and was associated with tumour aggressiveness and poor patient survival in human breast and rectal cancer[118,119]. Inhibition of EphA4 has been shown to overcome intrinsic resistance to chemotherapy[120]. Many of these other potential enhancer-hijacking targets do not have well-established roles in cancer pathogenesis, however, we did notice a number of genes involved in cell adhesion (*ALCAM, NCAM2, CADM2,* and *CDH9*) and 2 SLIT and NTRK like family members (*SLITRK1, SLITRK6*).

**Immunoglobulin translocations implicate distinct CNSL subtypes.** IG translocations are established oncogenic drivers of many lymphatic neoplasms[121–123]. *IGH-BCL6* fusions are recurrent in PCNSL[24], which mirrors observations of ABC-DLBCL[124]. *IGH-BCL2* fusions are more prominent in GCB-DLBCL[125]. We investigated the recurrent translocations (≥2 patients) in our cohort and identified five CNSL samples with *IGH-BCL6* translocations (Fig. 5a and Supplementary Fig. 7a–d). We also identified three cases with *IGH-BCL2* translocations (Fig. 5b and Supplementary Fig. 7e, f) one in each of SCNSL, SCNSL-M, and PCNSL-M,

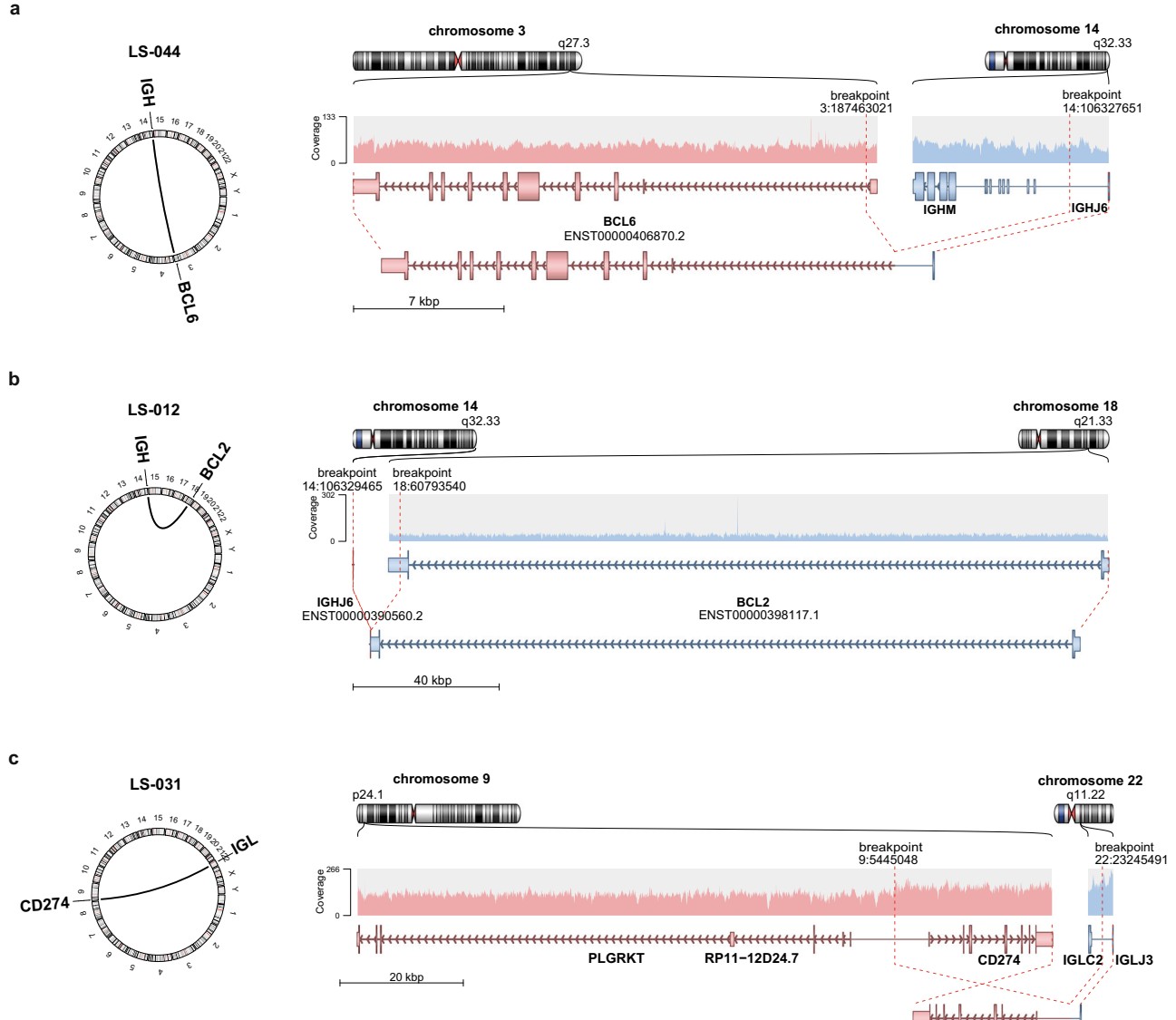

**Fig. 5 Immunoglobulin translocations in PCNSL.** Schematic representation of the translocation breakpoints involving *BCL6* (PCNSL patient LS-044; **a**), *BCL2* (SCNSL-M patient LS-012; **b**), and *CD274* (PD-L1; PCNSL patient LS-031; **c**). The left panels show a circular plot displaying the location of the translocation partners. The right panel shows the original and reconstructed translocation events. Shown are (from top to bottom) chromosome ideograms, read overage of the sites, the gene models, and the reconstructed translocation partners.

further implicating that meningeal and secondary CNSL are distinct from intraparenchymal PCNSLs. Two PCNSL cases showed *IGL* and *IGH* translocations with breakpoints close to *CD274* (PD-L1; Fig. 5c, Supplementary Fig. 7g), which resulted in strong PD-L1 protein expression (Supplementary Fig. 7h) and therefore implicates a potential target for immunotherapy. All other, non-recurrent translocations are listed in Supplementary Data 10.

The analyses of the IG breakpoints provided in all informative junctions evidence that these occurred due to illegitimate CSR or aberrant SHM, with the notable exception of the *IGH-BCL2* junctions, which were the consequence of an aberrant VDJ rearrangement (Supplementary Data 11). Thus, all IG translocations in PCNSL are supposed to occur in the GC process rather than in a pre B-cell.

**Mutational signatures in PCNSL.** Mutational signatures were analyzed with regard to SNVs (single base substitutions, SBS) and indels (ID) of all tumor samples as defined by

Alexandrov et al.[126] (Fig. 6). For single base substitution signatures (SBS) we found mutational patterns that have been associated with spontaneous deamination of 5-methylcytosine (SBS1), defective activity of the AID/APOBEC family (SBS2), failure of double-strand DNA break repair by homologous recombination (SBS3), SHM (SBS9), and damage by reactive oxygen species (SBS18). Additionally, the samples frequently revealed mutations caused by mutational signatures SBS5, SBS17b, and SBS40, which are of unknown etiology (Fig. 6a). The presence of SBS3, hallmark of defective DNA break repair by homologous recombination, and SBS40 may be therapeutically relevant as these indicate potential effectiveness of combination therapy with PARP inhibitors (e.g., Olaparib) alongside cytotoxic chemotherapy[127,128]. The three most prominent signatures in DLBCL, FL, and CNSL were SBS9, SBS5, and SBS40 (Fig. 6b). Direct comparison of PCNSL and DLBCL revealed that signature SBS1, which correlates with DNA replication at mitosis (mitotic clock)[126], was significantly enriched in PCNSL ($p = 0.0027$; Fig. 6c, Supplementary Fig. 8a–g).

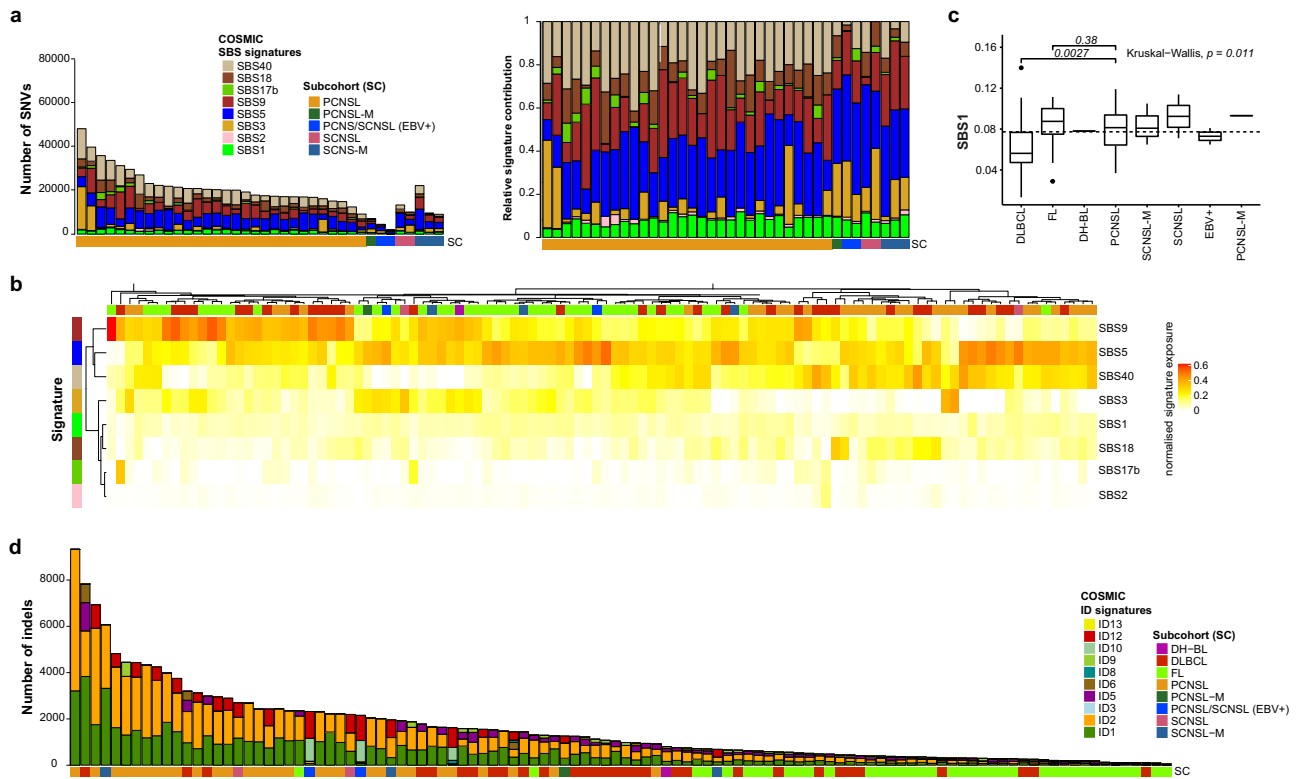

**Fig. 6 Mutational signatures in PCNSL.** Single base substitution (SBS) signature contribution in PCNSL and SCNSL. Stacked bar plots are ordered by subgroup and then decreasing mutations load (**a**). The left bar plot shows the number of SNVs and the right shows the normalized signature exposures. Each color corresponds to a mutations signature and the proportion of the color reflects the number or proportion of SNVs explained by a certain mutational signature. The heatmap shows the clustering pattern of the SBS mutational signatures in all CNSL and peripheral lymphomas (**b**), which revealed groupings mutational signatures SBS5, SBS9, and SBS40. Pairwise comparison of PCNSL with DLBCL or FL (Mann–Whitney U test) revealed that signature SBS1 was significantly enriched in PCNSL compared to DLBCL ($p = 0.027$; **c**). Box and whisker plots show median (center line), the upper and lower quartiles (the box), and the range of the data (the whiskers), excluding outliers. Small insertion and deletion (ID) signatures in CNSL and peripheral lymphoma (**d**) demonstrate elevated numbers of indels and mutational patterns associated with slippage during DNA replication of the replicated DNA strand (ID1) and template DNA strand (ID2) in PCNSL.

Analysis of small insertion and deletion signatures (ID) revealed mutational patterns associated with slippage during DNA replication of the replicated DNA strand (ID1) and template DNA strand (ID2); both of these signatures appeared significantly ($p < 1 \times 10^{-4}$, Wilcoxon) more prominent in PCNSL compared to DLBCL and FL (Fig. 6d), though different read-depths may have influence this analysis.

Interestingly, only CNSL samples but not DLBCL or FL revealed mutations caused by mutational signature ID12 that is of unknown etiology and has been observed in prostate adenocarcinoma and soft tissue liposarcoma[126].

**PCNSL RNA expression signatures are distinct from systemic DLBCL.** The relative rarity of PCNSL and limited availability of fresh frozen tissue have thus far complicated the implementation of larger molecular studies needed for patient stratification. To unravel the molecular signature of PCNSL, we employed an unsupervised consensus clustering approach (using the cola tool[129]) to identify expression groupings between PCNSL samples and samples from the ICGC MMML-seq project (mainly consisting of non-GCB and GCB type DLBCLs, and FLs). This yielded the following major clusters: FL, PCNSL, GCB-type DLBCL, ABC-type DLBCL, non-tumorous GC B-cells, and naïve B-cells (Fig. 7a). For each cluster, we identified signature gene sets that significantly correlated with the groupings. Interestingly, all meningeal PCNSL (PCNSL-M) and SCNSL-M grouped together

with either GCB- or ABC-DLBCL, clearly indicating that these subtypes are molecularly and pathomechanistically distinct from intraparenchymal CSNL, which formed one separate cluster suggesting a distinct signature of CNS tropism. The ABC-type DLBCL cluster was enriched for *MYD88* mutant samples, which were still distinct from *MYD88* mutant PCNSL at the gene expression level (Fig. 7a).

To further exclude an impact of potentially contaminating surrounding CNS tissue on gene expression signatures, we analyzed total RNA from normal brain controls ($n = 2$) and compared this to PCNSL. To investigate the gradient of various tumor cell contents of samples, we spiked increasing concentrations of RNA from non-diseased brain tissue into a PCNSL sample with very high tumor cell content (0, 20, 40, 60, and 80%). Then, we further stratified the PCNSL group by another round of consensus clustering using two different classification methods, which both revealed two groups (Fig. 7b and Supplementary Fig. 9a–c). The first PCNSL expression group (PCNSL subcluster 1) consisted of samples with high tumor cell content (determined by WGS and histopathological analysis). Expression of its signature gene set did not show similarity to normal brain tissue expression. However, the second PCNSL expression group (PCNSL subcluster 2) contained mainly samples with lower tumor cell content, and expression of its signature gene set was indeed similar to normal brain tissue expression (Fig. 7b). We identified the PCNSL signature gene sets relative to ABC and GCB type DLBCLs and FLs, and removed potential background signatures from contaminating brain tissue.

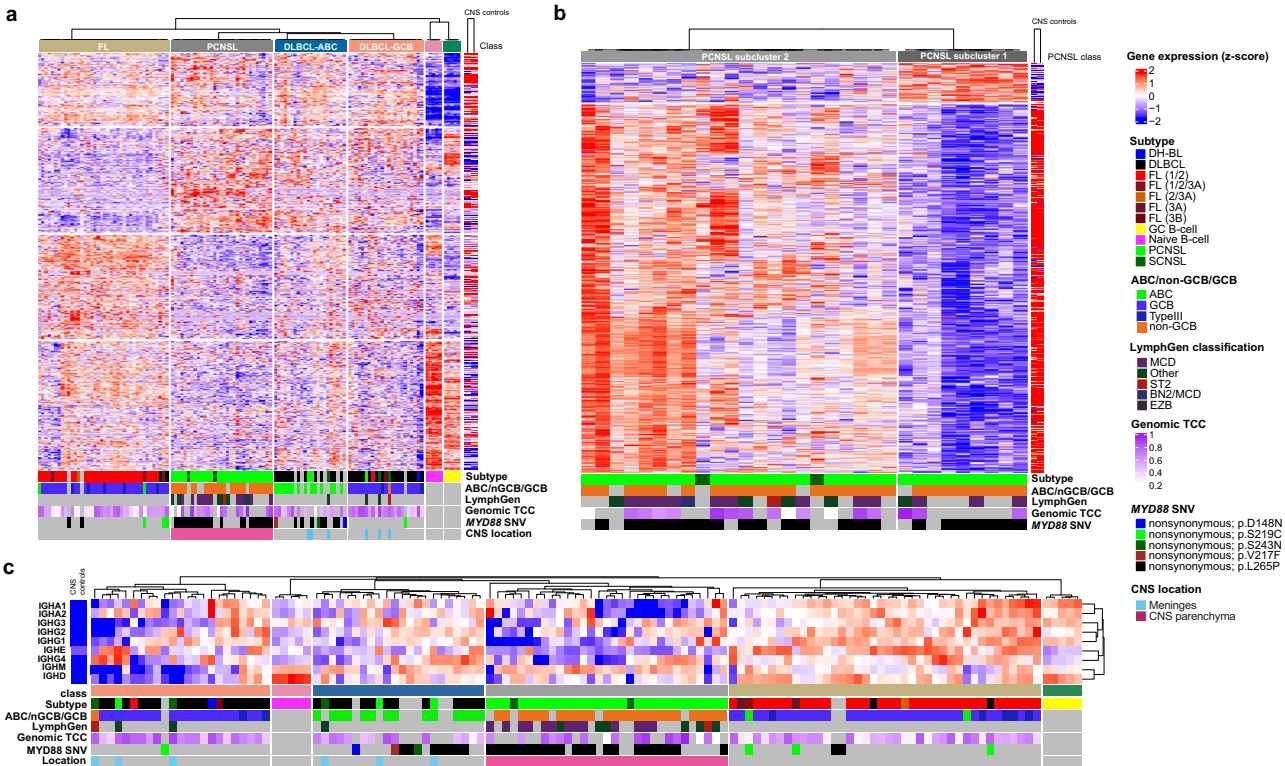

**Fig. 7 Transcriptomic signatures distinguish PCNSL from DLBCL.** Global gene expression patterns clearly separate PCNSL from other subtypes (ABC-DLBCL, GCB-DLBCL, and FL; **a**). The heatmap shows unsupervised consensus clustering of gene expression data. All SCNSL-M and PCNSL-M cluster with non-CNS-DLBCL, distinct from intraparenchymal PCNSL. The ABC-DLBCL cluster is enriched for *MYD88* mutant samples, which are distinct from *MYD88* L265P mutant PCNSL. Samples of normal germinal center (GC) and naïve B-cells were included as controls. Furthermore, normal brain tissue (CNS controls) samples were added to analyze the impact of normal brain tissue contamination at the RNA level. Consensus clustering (skmeans) with intraparenchymal PCNSL samples and CNS controls revealed two groups (**b**). The first PCNSL expression group (PCNSL subcluster 1, on the right) consisted of samples with high tumor cell content, the second (PCNSL subcluster 2, on the left) contained mainly samples with a lower tumor cell content, which signatures correlated well with normal brain tissue expression. PCNSL can be distinguished from ABC-DLBCL, GCB-DLBCL, and FL based on the expression of *IG* constant genes (**c**).

The marker genes in each group were identified based on differential gene expression analysis (Supplementary Data 12). Among the marker genes for PCNSL were e.g., *LAPTM5*, a CD40-related gene expressed in malignant B-cell lymphoma[130] and *ITGAE*, mediating cell adhesion, migration, and lymphocyte homing through interaction with E-cadherin[131]. We used Metascape[68] for functional analysis of marker genes. Further pathway and process enrichment analysis revealed that pathways such as 'ribonucleoprotein complex biogenesis', 'mRNA processing', 'cell cycle', 'RNA modification', 'DNA conformation change', and 'DNA-templated transcription, initiation' were enriched (Supplementary Fig. 9d). The top three level Gene Ontology biological processes included 'cellular component organization or biogenesis', 'metabolic process', and 'localization' (Supplementary Fig. 9e).

**Expression of IGHM is characteristic for PCNSL.** Additionally, we analyzed the expression of *IG* constant genes, which again revealed the same clusters as the unsupervised consensus clustering approach, demonstrating that PCNSL can be differentiated from DLBCL based on only the expression of *IG* constant genes. In contrast to DLBCL and FL, PCNSL show generally low expression of *IG* constant genes, but higher expression of *IGHM* (Fig. 7c).

**TERT expression but not telomere content upregulated in PCNSL.** Telomerase activity and telomerase reverse transcriptase (*TERT*) gene expression have been reported as prognostic factors

in PCNSL patients[132]. We used TelomereHunter, a software for detailed characterization of telomere maintenance mechanisms[133] to estimate the telomere content in a representative cohort of PCNSL, SCNSL, peripheral lymphoma, as well as non-tumorous naïve and GC B-cells as control[59]. In approximately 1/3 of the samples, the TCC-corrected telomere content was higher in the tumor than in the matched control (whole blood) (Fig. 8a and Supplementary Fig. 10a). Nevertheless, telomere content was not significantly different between the different histological, clinical and molecular subgroups irrespective of whether the results were corrected for TCC (Fig. 8b and Supplementary Fig. 10b (tumor/control log2 ratio)) or not (Supplementary Fig. 10c (uncorrected for the control sample)). As expected, we found a negative correlation between age and telomere content in the control (Supplementary Fig. 10d). However, expression of the *TERT* gene, the main activity of the encoded protein is the elongation of telomeres, was significantly higher in GC B-cells[59] and in PCNSL compared to ABC-DLBCL (Fig. 8c and Supplementary Fig. 10e). This was consistent with observations when stratifying samples by RNA subgroups, where *TERT* expression was significantly higher in PCNSL compared to ABC-DLBCL, GCB-DLBCL, and FL (Fig. 8d).

Interestingly, the higher *TERT* expression in PCNSL significantly correlated with normalized telomere content (uncorrected for the control sample: Pearson's $R = 0.67$, $p = 0.003$; (Fig. 8e) and telomere content T/C log2 ratio (Supplementary Fig. 10f)). However, SCNSL-M, DLBCL and FL did not show such trends

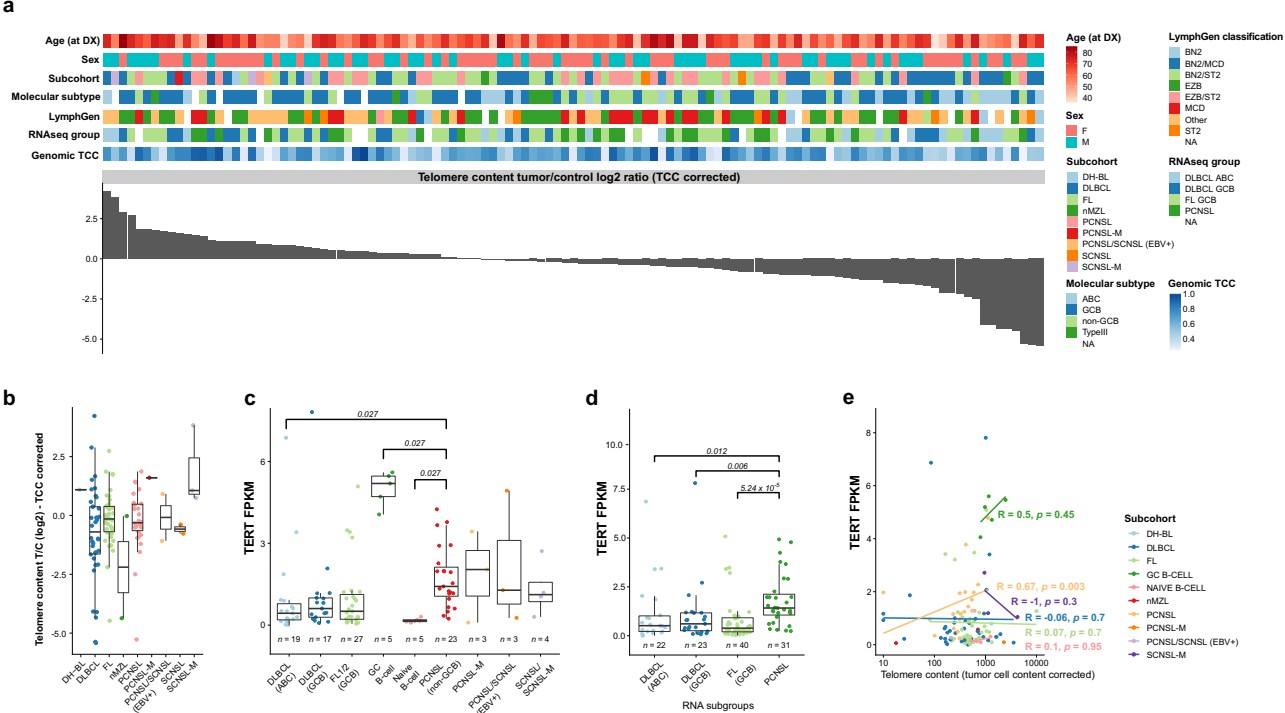

**Fig. 8 *TERT* expression correlates with telomere content in PCNSL.** Telomere content in DLBCL, FL, PCNSL, and SCNSL was estimated from WGS data using TelomereHunter. In about 1/3 of the samples, the telomere content was higher in the tumor than in the control sample (**a**). There was no statistical difference between PCNSL and other subcohorts in telomere content (**b**). TERT expression was significantly higher in PCNSL compared to subcohort ABC-DLBCL (**c**) and compared to RNA subgroups ABC-DLBCL, GCB-DLBCL, and FL (**d**). *P*-values in panels **c** and **d** are calculated using the one-sided Wilcox test adjusted for multiple testing using the Holm-Bonferroni method. Box and whisker plots show median (center line), the upper and lower quartiles (the box), and the range of the data (the whiskers), excluding outliers. A significant positive correlation between TERT expression and telomere content was only observed in PCNSL (**e**), using the Pearson correlation-coefficient test without correction for multiple testing.

($R = -1$, $p = 0.3$, $R = -0.06$, $p = 0.7$, and $R = 0.07$, $p = 0.7$, respectively (Fig. 8e)). This suggests that *TERT* has an active role in combatting telomere degradation in PCNSL. Two well-known promoter hotspot mutations (−124C>T (C228T) and −146C>T (C250T)) have been described to increase *TERT* expression and cell-cycle progression[134,135]. These mutations have been found in several solid and hematological malignancies including different brain tumors and PCNSL[136–138]. Therefore, we next investigated the *TERT* promoter mutation status in our WGS ($n = 38$) and FFPE extension cohort ($n = 31$). Sanger sequencing of the *TERT* promoter region was performed (i) for all WGS samples having only low coverage in the promoter sequence (below 40×, $n = 6$, Supplementary Fig. 10g), (ii) the FFPE extension cohort, and (iii) three oligodendrogliomas, known to carry high frequency *TERT* promoter mutation[139]. We detected no *TERT* promoter mutations in 67 samples of PCNSL and SCNSL (Supplementary Data 5, Supplementary Fig. 10h), while the well-known TERT rs2853669 polymorphism, which has been associated with increased cancer risk[140], was identified in 40% (14/35 (8 PCNSL, 6 SCNSL)) of the patients in the extension FFPE cohort. The Sanger sequencing results of two samples were not conclusive.

## Discussion

Here we have performed a comprehensive analysis of recurrent protein coding and non-coding mutations, CNVs, SVs, and driver mutations in a large cohort of PCNSL and compared the genetic features to systemic DLBCL and FL. The vast majority of PCNSLs are of non-GCB-DLBCL subtype[141] and share many genetic alterations with non-CNS ABC-DLBCL in the same signaling pathways. Previous studies made use of whole-exome sequencing[25,53] which (i) limits the investigation to protein-

coding regions and (ii) may not be ideal for understanding the patterns of mutational hotspots—e.g., attributed to AID induced SHM in B-cell non-Hodgkin lymphomas[142]—as well as the structural variation in genomes[143]. PCNSL showed significantly more SNVs and indels compared to systemic DLBCL, even in intronic and intergenic regions, also underlining the importance of non-protein coding aberrations in PCNSL pathogenesis. Many of the recurrent mutations in non-protein coding genes affected non-coding RNAs (ncRNAs), which are among other functions involved in epigenetic regulation of gene expression, cell differentiation, and development[82,144]. The molecular profile of SCNSL, on the other hand, corresponded to that of systemic DLBCL.

In line with previous results[25,43,145], we here demonstrate that PCNSL are defined by recurrent and often biallelic *CDKN2A* deletions, *MYD88* L265P mutations, and mutations that activate BCR signaling, genetic hallmarks of the DLBCL subtype MCD/C5[30,31]. Furthermore, we found high frequencies of SVs affecting the *IGH*, *IGL*, and *IGK* loci as well as losses of chromosome 6p affecting the *HLA* gene complex as a mechanism to escape recognition by cytotoxic T-cells[146]. *MYD88* L265P mutation and *CDKN2A* loss have been described as early mutational events in PCNSL[45] and we confirmed both to be major drivers in PCNSL. While *TP53* alterations seem to play a minor role in PCNSL, the *CDKN2A/B* genes encode several proteins that regulate either the p53 (p19 ARF) or the RB1 (p16 INK4a) pathway[147,148], underlining the relevance of the TP53 pathway in the context of PCNSL and cell cycle control.

The frequencies of *MYD88* mutations had varied between 38 and 94% in previous PCNSL studies[26,31,53,54,149], which might reflect a selection bias among small study populations, given the

rarity of PCNSL. This huge range could alternatively result also from an imprecise definition of PCNSL, which includes all malignant NHL within the brain, eyes, spinal cord, or leptomeninges without systemic involvement. In contrast, we here defined PCNSL as only intraparenchymal CNS-DLBCL and found a high prevalence of the *MYD88* L265P variant in this cohort (WGS cohort, extension FFPE cohort; mean: 73%). This is further supported by our robust classification of PCNSL by the RNA sequencing results, which demonstrated that the expression profiles of PCNSLs were distinct from PCNSL-M, SCNSL-M, and peripheral DLBCL without CNS manifestation, the latter three entities sharing similar profiles.

Moreover, SHM has previously been described as having a pathogenic role in PCNSL development and that its extent was greater there than in systemic DLBCL[80]. In agreement with previous reports, we identified several aSHM targets including the proto-oncogenes *PIM1*, *PAX5*, *BTG2*, and *OSBPL10*[25,53,80]. Exploiting a WGS approach, we observe additional mutational hotspots indicative of aSHM also in other genes including *MIR142*, *FHIT*, *ETV6*, *BTG1*, *GRHPR*, and *CD79B*. Our data suggest that katagis loci are reasonable indications of aSHM. We observed significantly higher RNA expression of genes with putative aSHM loci compared to those without. In addition, these putative aSHM loci were significantly enriched in genes involved in BCR signaling. Together this implicates that BCR signaling genes are both upregulated and targeted by putative aSHM, raising the question of cause and effect—is aSHM upregulating these genes, or is the high expression levels of these genes priming them for aSHM? This becomes even more complex when considering that highly expressed genes should have lower mutational rates due to transcriptional coupled repair[150].

The landscape of CNAs and SVs revealed potentially clinically exploitable deletion of *TOX* as a predictor for anti-PD1 response[103], amplification of *MALT1*, whose inhibition has been shown to be selectively toxic for ABC-DLBCL[151], and potential enhancer-hijacking events involving *PIK3C3* and *EPHA4*, whose inhibition has shown therapeutic advantage in a number of cancer models[113–115,120].

While the genetic landscape of PCNSL was described in some detail before[16–18,20,24,25,45,53,55], studies investigating the global gene expression profile of PCNSL have been scarce so far. Therefore, we performed RNA sequencing of 37 CNSL samples and 2 normal brain controls. Global gene expression profiles demonstrates that PCNSL are indeed distinct and can be distinguished from systemic ABC-DLBCL. This was perfectly mirrored based on the expression repertoire of *IG* constant genes, implicating the role of B-cell maturation in classification of PCNSL and other lymphomas, as employed in leukemia and multiple myeloma[152,153].

PCNSL are highly proliferative[154]. *TERT* activation confers unlimited proliferation, and activating *TERT* promoter mutations are frequent in different types of human cancers[155]. Mutations at two hotspots positions (−124G>A and −146G>A) are causal for enhanced *TERT* promoter activity. Bruno et al. have previously reported these *TERT* promoter mutations to be present in PCNSL located in the splenium[136]. Therefore, we investigated 69 CNSL (including 49 PCNSL), but could not identify any *TERT* promoter mutations suggesting that this mechanisms of *TERT* activation is likely not relevant in PCNSL. Yet, we observed significantly more *TERT* expression in PCNSL compared to non-CNS ABC-DLBCL and this was consistent when stratifying the cohort based on our RNAseq subgrouping. However, we were not able to identify increased telomere content in PCNSL (or MCD) compared to the other groups, suggesting a role of telomere maintenance to overcome telomere shortening, which is imposed by the high levels of proliferation. This concept was supported by a within-group correlation of *TERT* expression and normalized telomere content (Pearson's $R = 0.67$, $p = 0.0023$), implicating a role of *TERT* in overcoming telomere degradation in PCNSL. Supporting the high proliferation in these tumors, PCNSL showed significantly elevated presence of the mutational signatures SBS1, which correlates with DNA replication, as well as of ID1 and ID2 which are associated with slippage during DNA replication.

With our study, we have substantiated the genomic and transcriptomic alterations characterizing PCNSL. We show that PCNSL can be clearly distinguished from systemic DLBCL, having distinct expression profiles, *IG* expression, and translocation patterns, as well as specific combinations of genetic alterations that are characterized by genomic instability, BCR activation, and most importantly, oncogenic TLR and NFκB signaling, which should be in the focus of future drug development.

## Methods

**CNS lymphoma (CNSL) study cohort**. All procedures performed in this study were in accordance with the ethical standards of the respective institutional research committees and with the 1964 Helsinki declaration and its later amendments or comparable ethical standards. The ethics committee of the Medical Faculty Heidelberg and the Charité ethics committee (Charitéplatz 1, 10117 Berlin, Germany, approval number: EA1/245/13) approved the study. Informed consent was obtained from all participants in the study. Fresh frozen and paraffin-embedded PCNSL and SCNSL tumor tissue and matching blood samples (germline control) were acquired from the Department of Neuropathology, Charité, Berlin (Germany), and the Department of Neurosurgery, Heidelberg (Germany) from chemotherapy-naïve patients. Age at diagnosis, tumor localization, peripheral manifestation (bone marrow biopsy result, CT/MRI scan), first line therapy, as well as overall survival (OS) in months were evaluated. Additional control samples from age-matched, postmortem, and non-neoplastic brain ($n = 2$) were analyzed. The diagnosis was confirmed by at least two experienced (neuro)pathologists. The morphologic characteristics were assessed by using the fresh-frozen (FF) as well as the formalin-fixed and paraffin-embedded (FFPE) tissue sections of the respective tumor specimen. The tumor cell content in the cryopreserved sample material was estimated to be at least 60% based on histomorphological evaluation. Immunophenotypic characterization was performed on FFPE tissue sections (Supplementary Fig. 1b) of each tumor biopsy using an immunohistochemical panel including antibodies directed against CD20, CD10, BCL6, CD3, Ki67, MUM1/IRF4, and EBV (LMP1). To further exclude an EBV association, all cases with unclear EBV immunohistochemistry ($n = 9$) were investigated by an EBV-specific PCR as previously described[156] (Supplementary Fig. 1c, see "Methods"). For classifying GCB or non-GCB types, all samples were stratified according to the Hans classification[12] (CD10, BCL6, MUM1). We enrolled CNSL from a total of 51 patients for whole-genome (WGS, $n = 38$) and RNA sequencing (RNAseq, $n = 37$), including $n = 24$ samples subjected to both workflows. The study cohort and sample size as well as the experimental design, analysis workflow, diagnosis, and quality metrics of WGS and RNAseq are displayed in Fig. 1 and Supplementary Data 1. We included DLBCL confined to the CNS as PCNSL according to the recent WHO classification of tumors of haematopoietic and lymphoid organs and one of the tumors of the central nervous system. DLBCL, which presented initially with systemic, non-CNS, or synchronous systemic and CNS involvement were included as SCNSL[1,2,11,61,62]. In our SCNSL cohort, three patients presented with initial lymph node manifestation, one patient with testicular involvement, and three patients with involvement of parotid gland, liver, or urinary tract, respectively.

**ICGC MMML-Seq Consortium samples**. For comparison, we used and reanalyzed an early release of meanwhile published whole-genome and RNA sequencing data obtained by the ICGC MMML-Seq Consortium from systemic diffuse large B-cell lymphoma (DLBCL, total: $n = 36$, WGS: $n = 29$, RNAseq: $n = 36$, both workflows: $n = 29$), follicular lymphoma (FL, total: $n = 39$, WGS: $n = 39$, RNAseq: $n = 38$, both workflows: $n = 38$), and one "double hit" (DH)-lymphoma with a molecular BL signature[34]. In addition, we included WGS and RNAseq data from a single EBV-PCNSL case as well as RNAseq data from two nodal marginal zone lymphomas (nMZL) as well as naïve ($n = 5$) and GC B-cells ($n = 5$) as normal controls[157]. These data were obtained by the ICGC MMML-Seq consortium in accordance to protocols previously published[59,60].

**DNA and RNA isolation**. DNA and RNA were obtained from fresh frozen CNSL tumor samples. RNA and genomic DNA were isolated from 15 to 30 10 μm cryosection slices (depending on the tissue size). DNA from tumor samples and their matched blood controls was isolated according to standard procedures. Total RNA from tumor samples was extracted using the RNeasy® Plus Mini Kit (Qiagen, Hilden, Germany) according to the manufacturer's instructions. The RNA integrity

number (RIN) was determined using an Agilent 4200 TapeStation system (Agilent Technologies, Santa Clara, CA).

## Whole-genome sequencing and data processing

*Whole-genome sequencing.* The DNA libraries of the tumor and matched control samples were prepared according to the Illumina TruSeq Nano DNA Library protocol using the TruSeq Nano DNA library Preparation Kit (Illumina, Hayward, CA; estimated insert size of 350 bp). Paired-end sequencing was performed on Illumina HiSeq X (2 × 150 bp) instruments using the TruSeq SBS kit, Version 3.

*Alignment of sequencing reads.* Sequencing reads were aligned using the DKFZ alignment workflow from ICGC Pan-Cancer Analysis of Whole Genome projects (DKFZ AlignmentAndQCWorkflows v1.2.73, https://github.com/DKFZ-ODCF/AlignmentAndQCWorkflows). Briefly, read pairs were mapped to the human reference genome (build 37, version hs37d5) using bwa mem (version 0.7.8) with minimum base quality threshold set to zero [-T 0] and remaining settings left at default values[158], followed by coordinate sorting with biobambam bamsort (version 0.0.148) with compression option set to fast (1) and marking duplicate read pairs with biobambam bammarkduplicates with compression option set to best (9)[159]. To allow the required and meaningful comparability to previous whole genome sequencing studies in lymphomas[34,95,160,161], the human reference genome version GRCh37/hg19 was used.

*Small mutation calling and annotation.* Somatic small variants (SNVs and indels) in matched tumor normal pairs were called using the DKFZ in-house pipelines (SNVCallingWorkflow v1.2.166-1, https://github.com/DKFZ-ODCF/SNVCallingWorkflow; IndelCallingWorkflow v1.2.177, https://github.com/DKFZ-ODCF/IndelCallingWorkflow) as previously described[162]. Briefly, the SNVs were identified using samtools and bcftools version 0.1.1957[163] and then classified as somatic or germline by comparing the tumor sample to the control, and later assigned a confidence which is initially set to 10, and subsequently reduced based on overlaps with repeats, DAC blacklisted regions, DUKE excluded regions, self-chain regions, segmental duplication records as introduced by the ENCODE project[164] and additionally if the SNV exhibited PCR or sequencing strand bias. Only SNVs with confidence 8 or above were considered for further analysis. Tumor and matched blood samples were analyzed by Platypus[165] to identify indels. Indel calls were filtered based on Platypus internal confidence calls, and only indels with confidence 8 or greater were used for subsequent analysis. In order to remove recurrent artifacts and misclassified germline events, somatic indels that were identified as germline in at least two patients in the CNS lymphoma cohort were excluded.

The protein coding effect of somatic SNVs and indels from all samples were annotated using ANNOVAR[166] according to GENCODE gene annotation (version 19) and overlapped with variants from dbSNP10 (build 141) and the 1000 Genomes Project database. Mutations of interest were defined as somatic SNV and indels that were predicted to cause protein coding changes (non-synonymous SNVs, gain or loss of stop codons, splice site mutations, and both frameshift and non-frameshift indels), and also synonymous exonic mutations on non-coding genes.

*Tumor in normal contamination detection.* We applied the TiNDA (tumor in normal detection algorithm) workflow to account for potential tumor in normal contamination leading to false negative calls as previously described[162]. The TiNDA algorithm is implemented in the DKFZ indel calling workflow (v1.2.177, https://github.com/DKFZ-ODCF/IndelCallingWorkflow). Briefly, the B-allele frequency (BAF) was calculated from the tumor and control samples. Positions overlapping with common variants were filtered out. Then, the clustering algorithm from Canopy[167] was applied to the BAF values for the positions in tumor vs control using a single pass run, assuming 9 clusters. The clusters that were determined to be tumor-in-normal had to have 75% of positions above the identity line (where the VAF in the tumor sample is the same as the VAF in the control sample). These identified mutations were then reclassified as somatic instead of the original germline annotation. All but 4 CNSL WGS samples exhibited evidence for tumor in normal. On average 31 SNVs (range 0–136) were "rescued" in PCNSL, 6 in PCNSL-M (6-6, single sample), 27 (19–34) in SCNSL, 22 (9–43) in SCNSL-M, and 0 in the EBV-positive sub-cohorts (0, 2 samples). In total, only 6 SNVs with protein coding effects were rescued, including the MYD88 p.L265P mutation in sample LS-0102, which had 3 of 47 read support in the control, and 86 of 170 reads supporting the variant in the tumor sample (Supplementary Data 13). In our series, we only found a very low level of tumor in normal. The rescued mutations followed a genomic distribution similar to the overall mutational landscape of PCNSL. We observed that rescued mutations were 1% exonic (compared to 1% in the mutational landscape), 32% intronic (c.f. 33%), 53% intergenic (c.f. 55%), and 13% on ncRNA (c.f. 12%).

*Genomic structural rearrangements.* Genomic structural rearrangements (SVs) were detected using SOPHIA v.34.0[168] as implemented in the DKFZ structural variation calling workflow (SophiaWorkflow:1.2.16, https://github.com/DKFZ-ODCF/SophiaWorkflow). Briefly, SOPHIA uses supplementary alignments as produced by bwa-mem as indicators of a possible underlying SV. SV candidates are filtered by comparing them to a background control set of sequencing data obtained using normal blood samples from a background population database of 3261 WGS samples from patients from published ICGC-PedBrain, ICGC-MMMLseq and ICGC-Prostate studies and DKFZ-HIPO studies, sequenced using Illumina HiSeq 2000, 2500 (100 bp) and HiSeq X (151 bp) platforms and aligned uniformly using the same workflow as in this study. All studies have been approved by appropriate ethics committees. Gencode V19 was used for the gene annotations. We used the script draw_fusions.R from the Arriba package[169] to visualize SVs generated by SOPHIA.

*Copy number alterations and allelic imbalances.* Allele-specific copy-number aberrations were detected using ACEseq (allele-specific copy-number estimation from WGS)[170] as implemented in the DKFZ CNV calling workflow (ACEseqWorkflow:1.2.8-4, https://github.com/DKFZ-ODCF/ACEseqWorkflow). ACEseq determines absolute allele-specific copy numbers as well as tumor ploidy and tumor cell content (TCC) based on coverage ratios of tumor and control as well as the B-allele frequency (BAF) of heterozygous SNPs. SVs called by SOPHIA were incorporated to improve genome segmentation.

Final copy number segments were further smoothed to calculate the total number of gains and losses. Neighboring segments were merged if they rounded to the same copy number and deviated by less than 0.5 copies in case of segments <20 kb or deviated by less than 0.3 copies otherwise. Remaining segments <500 kb were merged with their closer neighbor based on allele-specific and total copy number and once again segments smaller than 2 Mb deviating by less than 0.4 copies were merged. Based on the resulting segments the number of gains and losses was estimated.

Furthermore, the fraction of aberrant genome was calculated as the fraction of the genome that is classified either as duplication or deletion (>0.7 deviation from the ploidy) or was identified as a loss of heterozygosity.

*Classification of mutational hotspots (kataegis events).* Mutational hotspots indicating putative kataegis events (likely due to SHM or aberrant SHM (aSHM)) were defined as regions with at least 6 somatic SNVs within an average intermutational distance of 1000 bp or less, as previously used by Alexandrov and colleagues[90]. A gene was described to be targeted by kataegis if its definition (from Gencode version 19 gene models) overlapped with at least 1 kataegis region in at least 1 sample. While many of these kataegis loci are indeed SHM/aSHM targets, located 2.5 kb from the transcription start site (TSS), we cannot completely control for all PCNSL-specific TSSs due to the normal brain background tissue.

*Mutational signatures.* Supervised mutational signature analysis was performed using YAPSA development version 3.13[171] using R 4.0.0. Briefly, the linear combination decomposition (LCD) of the mutational catalog with known and pre-defined PCAWG COSMIC signatures[126] was computed by non-negative least squares (NNLS). The mutational signature analysis was applied to the mutational catalogs for SNVs (or single base substitutions, SBS) and indels of all tumor samples. Signature-specific cutoffs were applied and cohort level analysis was used for detecting signatures as recommended by Huebschmann et al.[34]. The cutoff used corresponds to "cost factors" of 10 for SNVs and 3 for indels in the modified ROC analysis.

*Integration of different variant types.* SNVs, indels, SVs, and CNAs were integrated in order to account for all variant types in the recurrence analysis. All genes with SNVs or indels in coding regions (nonsynonymous, stop gain, stop loss, splicing, frameshift, and non-frameshift events) and ncRNA (exonic) were included. Any SV with breakpoints directly lying on a gene (SV direct) were considered for oncoprints, however, SVs were also annotated to a gene when they were either within 100 kb of a gene (SV near), or to the closest gene (SV close) for SV recurrence analysis to account for regulatory mutations such as enhancer hijacking events. Genes were annotated with CNAs if they were completely or partially affected. Chromosome level CNVs events were determined once >30% of a chromosome arm was altered. Only focal CNA events were taken into account for variant integration, as these are more likely to target specific genes within the affected region than large events such as whole chromosome arm events. To capture the precise target of CNVs, we employed results from GISTIC. Finally, genes affected by SNVs, indels, directly hit by SVs, or genes with focal CNAs were considered for the recurrence analysis and added to the oncoprints. The mutations were integrated and plotted as oncoprint plots using using R v3.4.0 (library yapsa v3.13), perl v5.26.2 (libraries perl-getopt-long v2.50) and bedtools v2.16.2. SV cohort plots were generated using perl v5.20.0, bedtools v2.24.0, R v3.3.1 (libraries circlize v0.4.5 and dplyr v0.7.8), using the gencode v19 gene models for annotation.

*Mutual exclusivity and inclusivity analysis.* Mutual exclusivity analysis was performed to investigate the relationship between *MYD88* mutations with other implicated drivers from the IntOGen analysis including SNVs, indels, SVs, CNAs.

The minimal recurrence threshold was set to 5. We applied the commonly used Fisher's exact test and the CoMET test[172] (v0.1.5) for both co-occurrence and mutual exclusivity using R (v3.4.0). Fisher's right tailed test was used to support co-occurrence when the number of samples with alterations in both genes is significantly higher than expected by chance. Additionally, Fisher's left tailed test was used to suggest mutual exclusivity when the number of samples with alterations in both genes is significantly lower than expected. Resultant p-values were corrected for multiple testing by FDR.

*Mutational significance analysis.* The IntOGen pipeline[173] algorithm was applied to identify significant cancer drivers in the core set of PCNSL samples (n = 30) based on the hg19 genome assembly. IntOGen v 3.0.8 was installed via conda from the bbglab anaconda channel. The relevant conda environment setup included explicit definitions of python v3.5.0 (with libraries scipy v0.16.0, pycurl v7.43.0.3, numpy v1.10.0, pandas 0.17.0). In addition, a local installation of perl v5.20.3.1 (with libraries perl-digest-perl-md5 v1.9, perl-threaded v5.26.0) was used, with installation of perl libraries Digest-MD5 v2.52 via cpan and perl-DBI v1.626 via yum package managers. The background intogen database (bgdata) was automatically downloaded using the command 'intogen -setup' which downloaded the 20150729 background databases. The IntOGen run specific parameters included running on 4 cores, Matlab Compiler Runtime v8.1 (2013a) and MutSigCV v1.4. Significance thresholds of 10% FDR were used for oncodrivefm, oncodriveclust and mutsig. Sample thresholds of 2 and 5 were used for oncodrivefm and oncodriveclust respectively. IntOGen reported 50 genes to be significant drivers.

Significant CNV were identified using GISTIC v2.0.23 using MCR v83, using the following parameters: "-broad 1 –genegistic 1 -savegene=1 -brlen=0.8 -conf=0.2 -maxseg=2500".

*Telomere content estimation.* The telomere content was determined from WGS data using the software tool TelomereHunter[59] (v1.1.0) which uses python v3.5.6 (using libraries pyyaml v3.13, pysam v0.9.1, pynacl v1.2.1), samtools v1.3.1, bcftools v1.3.1 and htslib v1.3.2. Telomere hunter was run using default settings (filtering of telomere reads: at least 6 telomere repeats per 100 bp read length)[133]. Briefly, unmapped reads or reads with a very low alignment confidence (mapping quality lower than 8) containing six non-consecutive instances of the four most common telomeric repeat types (TTAGGG, TCAGGG, TGAGGG, and TTGGGG) were extracted. The telomere content was determined by normalizing the telomere read count to all reads in the sample with a GC-content of 48–52%. In the case of tumor samples, the telomere content was further corrected for the tumor cell content (TCC, as estimated by ACEseq) using the following formula as previously described[59], which corrects for inter-patient differences in telomere content assuming that the non-malignant cells in the tumor sample have a similar telomere content as in the control sample, as shown in Eq. 1:

$$T_{TCCcorrected} = (T - C(1 - TCC))/TCC \quad (1)$$

Here, T and C are the telomere contents of the tumor and control sample, and $T_{TCCcorrected}$ is the TCC-corrected telomere content of the tumor sample.

*LymphGen classification.* All WGS samples were classified according to the LymphGen v2.0 algorithm described by Wright et al.[31] which categorizes DLBCL samples into the different genetic subtypes MCD, N1, A53, BN2 ST2, EZB (MYC + and MYC−), based on genetic aberrations in subtype predictor genes. The algorithm requires information on mutations, copy-number alterations, and fusions. The results of the DKFZ SNV and indels calling workflows were used to define the small mutations. BCL2 and BCL6 translocations were determined by Sophia calls, and copy number changes were derived from the DKFZ CNV workflow (ACEseq) results. The outputs from all workflows were filtered for somatic regions with all different variations occurring in exons and 5′UTR region of the gene. The files were created using Python and Perl scripts based on the description provided on the LymphGen website [https://llmpp.nih.gov/lymphgen/LymphGenInstructions.pdf?v=1600863825]. The individual sample inputs are further merged together to form the input dataset for the LymphGen algorithm and uploaded to the website [https://llmpp.nih.gov/lymphgen/lymphgendataportal.php] for classification the samples.

According to Wright et al.[31], the results are displayed in Supplementary Data 1. Samples, where only RNA was available, were listed as "NA" in Supplementary Data 1.

## RNA sequencing and data processing
*RNA library preparation and sequencing.* RNA libraries of the tumor samples and normal brain samples were prepared using the TruSeq RNA library preparation Kit Set A and B, following the manufacturer's instructions at an insert size of ~300 bp. Two barcoded libraries were pooled per lane and sequenced on Illumina HiSeq2000 or HiSeq4000 platforms.

*RNAseq alignment and expression quantification.* RNAseq reads were aligned and gene expression quantified using the DKFZ RNAseq workflow (v1.2.22-6, https://github.com/DKFZ-ODCF/RNAseqWorkflow) as previously described[174]. Briefly, the RNAseq read pairs were aligned to the STAR index generated reference genome (build 37, version hs37d5) using STAR in 2 pass mode (version

2.5.2b)[174,175]. Duplicate reads were marked using sambamba (version 0.6.5) and BAM files were coordinate sorted using SAMtools (version 0.1.19). featureCounts (version 1.5.1)[176] was used to perform non-strand-specific read counting for genes over exon features based on the Gencode V19 gene model (without excluding read duplicates). When both read pairs aligned uniquely (indicated by a STAR alignment quality score of 255) they were used towards gene reads counts. For total library abundance calculations, during TPM and FPKM expression values estimation, genes on chromosomes X, Y, MT, and rRNA and tRNA were omitted as they can introduce library size estimation biases.

Hierarchical consensus clustering was applied using the cola package (version 1.5.6) with "MAD" as top-value method and "kmeans" as partitioning method. Classification on CNS samples was applied using cola with "ATC" as top-value method and "skmeans" as partitioning method. All other parameters took default values[129].

*RNA dilution experiment.* To further investigate the impact of brain tissue contamination in unsupervised clustering analysis of gene expression data on PCNSL, we performed a serial dilution experiment with total RNA from a PCNSL sample considered "pure" (LS-027, estimated tumor cell content >80%) and a normal brain tissue control (CTRL). Total RNA from LS-027 was mixed with CTRL RNA with increasing concentrations (0, 20, 40, 60, and 80%) and sequenced. The z-score transformed TPM expression levels for PCNSL subcluster 1 and subcluster 2 signature genes for the serially diluted H050-0027 sample was compared against the cohort and individually using clustering analysis.

*Differential expression (DE) analysis to identify signature genes.* DE of genes was analyzed using DESeq2 (version 1.14.1) with default settings using raw read counts from featureCounts. Genes without any count in all samples were excluded from the analysis.

## Validation of whole-genome and RNA sequencing results
*Sanger sequencing.* Bidirectional Sanger sequencing (bSS) was performed (i) to validate WGS results in the PCNSL/SCNSL study cohort if sufficient DNA quantity was available (total: n = 35; PCNSL: n = 26; PCNSL-M: n = 1; SCNSL: n = 2; SCNSL: n = 3; EBV+: n = 2), and (ii) to identify mutations of recurrently mutated candidates in a larger set of additional PCNSL/SCNSL FFPE samples (FFPE extension cohort). The following genes were analyzed: MYD88, KMT2D (MLL2), HLA-B, SETD1B, HIST1H1E, CD79B, BTG1, MYC, TP53, TERT, GRHPR, TBL1XR1, DST, PRDM15, OBSCN, FAT4, GRP98, and OSBPL10. Briefly, the PCR conditions were: 94 °C for 4 min (1 cycle), followed by 3 cycles of 94 °C for 30 s, 61 °C for 45 s, 72 °C for 60 s, 3 cycles of 94 °C for 30 s, 59 °C for 45 s, 72 °C for 60 s, 3 cycles of 94 °C for 30 s, 57 °C for 45 s, 72 °C for 60 s, 31 cycles of 94 °C for 30 s, 55 °C for 45 s, 72 °C for 60 s, and finally extension at 72 °C for 10 min with AmpliTaq™ 360 DNA Polymerase (Applied Biosystems, Waltham, USA). The PCR primers for the genomic regions of interest are displayed in Supplementary Data 14. Sequencing was performed at Eurofins Genomics, Ebersberg, Germany. 180/189 (95%) of the selected variants (allele frequency above 10%) identified by WGS were confirmed. The results are displayed in Supplementary Data 4 and 5.

*Formaldehyde-fixed paraffin-embedded (FFPE) CNSL extension cohort.* Candidate genes were validated in 31 additional FFPE specimens of PCNSL (n = 19), SCNSL (n = 9), and n = 3 EBV positive cases. The DNA was extracted using the QIAamp DNA FFPE Tissue Kit (Qiagen, Hilden, Germany) according to the manufacturer's instructions. The following, recurrently mutated genes (exons) were investigated: PIM1 (exons 1–4), MYD88 (exon 5), GPHPR (exons 2, 4, and 5), TBL1XR1 (exons 7, 8, 10, 12, and 14), KMT2D (exons 32, 34, 38, 48, and 50), KLHL14 (exon 2), HLA-B (exons 2, 3), PRDM15 (exons 9–12), GPR98 (exons 65, 70, and 81), DST (exons 13, 14, 23, 24, and 36), OBSCN (exons 32, 63, 64, and 85), FAT4 (exons 1, 9, and 17), HIST1H3D (exon 2), HIST1H1E (exon 1), TERT (promoter region). The results are displayed in Supplementary Data 5.

*Fluorescence in situ hybridization (FISH).* FISH analysis was performed as previously described[177]. Briefly, 4 µm FFPE sections were deparaffinized, dehydrated, and incubated in pre-treatment solution (Dako, Denmark) for 10 min at 95–99 °C. Samples were treated with pepsin solution for 3–6 min at 37 °C, washed, dehydrated, air dried, and incubated with the respective DNA probe: CDKN2A (9p21.3): Orange, Biocare Medical, USA; Vysis CEP 9 SpectrumGreen Probe (Abbott), The Netherlands). The sections were sealed, denatured in humidified atmosphere at 82 °C for 5 min, and then incubated overnight at 45 °C to achieve hybridization. After post-hybridization washing, slides were counterstained with 4′6-diamidino-2-phenylindole (DAPI) and analyzed using an automated scanning system (Duet, BioView Ltd. Rehovot, Israel; Supplementary Fig. 1d).

*Real-time quantitative PCR (RT-qPCR).* We performed SYBR Green quantitative real-time PCR (qPCR) measuring six amplicons covering the CDKN2A/B gene (Supplementary Fig. 1e) as well as five amplicons covering the FHIT, NOTCH4, SPIB, and MIR650 gene. The primer sequences are annotated in Supplementary

Data 14. qPCR analysis was performed on ABI Prism 7900HT Sequence Detection System (Applied Biosystems, Foster City, CA, USA).

**Immunohistochemical (IHC) procedures**. Immunohistochemical stainings were performed on a Benchmark XT autostainer (Ventana Medical Systems, Tuscon, AZ, USA) with standard antigen retrieval methods (CC1 buffer, pH8.0, Ventana Medical Systems, Tuscon, AZ, USA) using 7 μm-thick frozen or 4 μm-thick FFPE tissue sections. The following primary antibodies were used: monoclonal mouse anti-BCL6 (DAKO, M7211, 1:10), monoclonal mouse anti-CD10 (Novocastra, NCL-CD10-270, 1:10), monoclonal mouse anti-CD20 (DAKO, M0755, 1:400), polyclonal rabbit anti-CD3 (DAKO, A0452, 1:100), monoclonal mouse anti-CD45 (DAKO, M0701, 1:400), monoclonal mouse anti-CD79a (DAKO, M7051, 1:100), monoclonal mouse anti-EBV-LMP1 (DAKO, M0897, 1:1000), monoclonal mouse anti-Ki-67 (DAKO, M7240, 1:100), monoclonal mouse anti-MUM1 (DAKO, M7259, 1:50), monoclonal mouse anti-PD-L1 (Cell Signaling, 13684, 1:200). The iVIEW DAB Detection Kit (Ventana Medical Systems, Tuscon, AZ, USA) was used according to the manufacturer's instructions. Sections were counterstained with hematoxylin, dehydrated in graded alcohol and xylene, mounted, and coverslipped. IHC stained sections were evaluated by two skilled neuropathologists with concurrence. The DLBCL subtypes of GCB and non-GCB were categorized using CD10, BCL6, and MUM1 according to the Hans classification[12].

**Epstein-Barr virus PCR**. EBV-specific PCR was performed as previously described[156]. Briefly, a highly conserved region of the EBNA-1 (BKRF1) gene specific for EBV was amplified by endpoint PCR using the following primers: 5′-GAG GGT GGT TTG GAA AGC-3′ and 5′-AAC AGA CAA TGG ACT CCC TTA G-3′, 0.1 μM each. The PCR conditions were: 95 °C for 5 min (1 cycle), followed by 40 cycles of 94 °C for 1 min, 55 °C for 2 min, 72 °C for 3 min, and finally extension at 72 °C for 7 min with ThermoPrime™ Taq DNA Polymerase (Thermo Fisher Scientific, Waltham, USA). Subsequently, amplification products were analyzed by ELISA (Roche, Basel, Switzerland).

**Statistics and reproducibility**. No statistical methods were used to predetermine sample sizes. We included all individuals with DLBCL of the CNS where sufficient material was available as specified in the description of study design. No data were excluded from the analyses. Statistical details for each analysis are mentioned in each figure legend or in the respective part of the text. WGS, RNA-sequencing, Sanger sequencing, quantitative real-time PCR, immunohistochemical stainings, and FISH were performed in a blinded fashion. Evaluation of histological and immunohistochemical stainings, as well as FISH images, was performed separately by at least two independent (neuro-)pathologists in Berlin and Heidelberg. Histological staining, immunohistochemistry, and FISH analyses were replicated at least once. CDKN2A/B FISH was performed exemplarily for $n = 4$ CNSL patients. The representative images shown were adjusted in brightness and contrast to different degrees (depending on the need resulting from the range of brightness and contrast of the raw images) in Adobe Photoshop, and for these cases, raw image files are publicly available [https://doi.org/10.5281/zenodo.6054242][92]. The experiments were not randomized.

**Reporting summary**. Further information on research design is available in the Nature Research Reporting Summary linked to this article.

## Data availability

The whole genome sequencing (WGS) and RNA sequencing data of the 51 CNSL samples generated in this study as well as the raw data of Sanger sequencing and quantitative real-time PCR data have been deposited in the European Genome-Phenome Archive under the accession number EGAS00001005339. The data are available under controlled access due to the sensitive nature of genome sequencing data, and access can be obtained by contacting the appropriate Data Access Committee listed for each dataset in the study. Access will be granted to commercial and non-commercial parties according to patient consent forms and data transfer agreements. We have an institutional process in place to deal with requests for data transfer. A response to requests for data access can be expected within 14 days. After access has been granted, the data is available for two years. Access to the ICGC MMML-Seq raw sequencing data is available via the EGA under the accession number EGAS00001002199 and EGAS00001001692. Access to the ICGC MMML-Seq data is available via the data access committee of the ICGC (www. ICGC.org). Raw image files of histological stainings, immunohistochemistry, and FISH images generated in this study, as well as all somatic mutation calls, integrated mutations tables, and RNAseq counts on which the analysis was performed have been deposited publicly at Zenodo [https://doi.org/10.5281/zenodo.6054242][92]. The uncropped PCR gel images as well as the processed real-time PCR data and Kaplan-Meier survival data shown in Supplementary Fig. 1 are provided in the Source Data file with this paper. The remaining data are available within the article, Supplementary Information or Source Data file.

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

## Acknowledgements

We thank the German Cancer Research Center (Deutsches Krebsforschungszentrum, DKFZ) Omics IT, Data Management Core Facility (ODCF), and the sequencing unit of the Genomics and Proteomics Core Facility (GPCF) for providing excellent technical support. We thank the NCT Molecular Precision Oncology Program for technical support and funding through project numbers HIPO H050 and A050 (SWi). We thank the German Cancer Consortium (DKTK), Partner site Berlin for technical support and funding through XD013 sequencing (JR, FH). Cartoon images in Fig. 1a, b were partially created with BioRender.com. We are indebted to Stefanie Mende, Petra Matylewski, Kathrein Permien, Vera Wolf, Sandra Meier, and Silvia Stefaniak for excellent technical assistance. We thank Werner Stenzel, Arend Koch, David Capper, Christine Sers and Ingeborg Tinhofer-Keilholz for valuable experimental advice. JR is a participant in the BIH-Charité Clinical Scientist Program funded by the Charité – Universitätsmedizin Berlin and the Berlin Institute of Health. CL is supported by postdoctoral Beatriu de Pinós from Secretaria d'Universitats I Recerca del Departament d'Empresa i Coneixement de la Generalitat de Catalunya and by Marie Sklodowska-Curie COFUND program from H2020 (2018-BP-00055). The ICGC MMML-seq Project has been supported by the German Ministry of Science and Education (BMBF) in the framework of the ICGC MMML-Seq (01KU1002A-J) and the ICGC DE-Mining (01KU1505G and 01KU1505E). Research of RS on PCNSL is supported by Deutsche Krebshilfe (grant: 70113053).

## Author contributions

J.R., R.K., E.S., F.P., S.B., D.L., and M.W. performed experiments and analyzed data. Z.G., L.S., D.H., U.H.T., X.P.H., N.P., M.S., C.L., D.J., G.P.B., S.S., S.U., and N.I. carried out the bioinformatics analyses and processed data. D.P., H.R., and A.O. performed sample acquisition and histological analyses. I.A. performed histological analyses. D.L. performed FISH analyses. A.K., M.M., J.O., K.F., P.V., D.M., Y.W., A.J., and A.U. clinically assessed or operated patients and contributed to sample acquisition. C.H.M., S.W.i., and J.R. coordinated sample acquisition. S.W.i. and S.W.o. oversaw sequencing of the samples. R.S., M.S., L.T., and R.K.ü. provided data from the MMML cohort. Experimental design and execution were overseen by F.L.H., S.W.i., O.W., C.H.M., B.B., C.A.S., R.S., M.H., R.K.ü., and R.E. J.R., N.I., S.W.i., F.L.H., and R.S. designed the study. RS coordinated the ICGC MMML-Seq network. J.R. and N.I. wrote the manuscript. J.R. prepared the final figures. F.L.H., S.W.i., O.W., C.H.M., B.B., C.A.S., R.S, and M.H. critically revised the manuscript. All authors critically read, reviewed, and approved the final manuscript.

## Funding

## Competing interests

The authors declare no competing interests.

## Additional information

[1]Department of Neuropathology, Charité - Universitätsmedizin Berlin, corporate member of Freie Universität Berlin, Humboldt-Universität zu Berlin, Berlin Institute of Health, Berlin, Germany. [2]German Cancer Consortium (DKTK), Partner Site Charité Berlin, Berlin, Germany. [3]Institute of Pathology, Universitätsmedizin Greifswald, Greifswald, Germany. [4]Digital Health Center, Berlin Institute of Health (BIH) at Charité - Universitätsmedizin Berlin, Berlin, Germany. [5]Molecular Precision Oncology Program, NCT Heidelberg, Heidelberg, Germany. [6]Division of Applied Bioinformatics, German Cancer Research Center (DKFZ), Heidelberg, Germany. [7]Division of Translational Medical Oncology, National Center for Tumor Diseases (NCT), Heidelberg, Germany. [8]Computational Oncology, Molecular Diagnostics Program, NCT Heidelberg, Heidelberg, Germany. [9]Heidelberg Institute for Stem Cell Technology and Experimental Medicine (HI-STEM gGmbH), Heidelberg, Germany. [10]Department of Pediatric Oncology, Hematology and Immunology, University Hospital, Heidelberg, Germany. [11]German Cancer Consortium (DKTK), Heidelberg, Germany. [12]Division Neuroblastoma Genomics, German Cancer Research Center (DKFZ), Heidelberg, Germany. [13]Institute of Human Genetics, Ulm University & Ulm University Medical Center, Ulm, Germany. [14]Institut d'Investigacions Biomèdiques August Pi Sunyer (IDIBAPS), CIBER de Cáncer, Barcelona, Spain. [15]Division of Molecular Genome Analysis, German Cancer Research Center (DKFZ), Heidelberg, Germany. [16]Hopp Children's Cancer Center, NCT Heidelberg (KiTZ), Heidelberg, Germany. [17]Division of Pediatric Neurooncology, German Cancer Consortium (DKTK), German Cancer Research Center (DKFZ), Heidelberg, Germany. [18]Bioinformatics and Omics Data Analytics, German Cancer Research Center (DKFZ), Heidelberg, Germany. [19]Biomedical Informatics, Data Mining and Data Analytics, Augsburg University, Augsburg, Germany. [20]Institute of Cell Biology (Cancer Research), Medical Faculty, University of Duisburg-Essen, Essen, Germany. [21]German Cancer Consortium (DKTK), Partner Site Essen, Essen, Germany. [22]Department of Hematology, Oncology and Cancer Immunology, Charité - Universitätsmedizin Berlin, corporate member of Freie Universität Berlin, Humboldt-Universität zu Berlin, Berlin Institute of Health, Berlin, Germany. [23]Department of Neurosurgery, Charité - Universitätsmedizin Berlin, corporate member of Freie Universität Berlin, Humboldt-Universität zu Berlin, Berlin Institute of Health, Berlin, Germany. [24]Department for Neurosurgery, Vivantes Klinikum im Friedrichshain, Berlin, Germany. [25]Department for Neurosurgery, Vivantes Klinikum Neukölln, Berlin, Germany. [26]Department of Hematology and Oncology, Georg-Augusts-University of Göttingen, Göttingen 37075, Germany. [27]Department of Pathology, Charité - Universitätsmedizin Berlin, corporate member of Freie Universität Berlin, Humboldt-Universität zu Berlin, Berlin Institute of Health, Berlin, Germany. [28]Medical Department of Hematology and Oncology, Johannes Kepler University, Kepler Universitätsklinikum, Linz, Austria. [29]Department of Hematology, Oncology and Tumor Immunology and Molekulares Krebsforschungszentrum, Charité - Universitätsmedizin Berlin, corporate member of Freie Universität Berlin, Humboldt-Universität zu Berlin, Berlin Institute of Health, Berlin, Germany. [30]German Cancer Research Center (DKFZ), Heidelberg, Germany. [31]Genomics and Proteomics Core Facility, German Cancer Research Center (DKFZ), Heidelberg, Germany. [32]Department of Neurosurgery, University Hospital Heidelberg, Heidelberg, Germany. [33]Cluster of Excellence, NeuroCure, Charitéplatz 1, 10117 Berlin, Germany. [34]German Center for Neurodegenerative Diseases (DZNE) Berlin, 10117 Berlin, Germany. [68]These authors contributed equally: Josefine Radke, Naveed Ishaque. [69]These authors jointly supervised this work: Reiner Siebert, Stefan Wiemann, Frank L. Heppner. *A list of authors and their affiliations appears at the end of the paper. ✉email: josefine.radke@med.uni-greifswald.de; naveed.ishaque@bih-charite.de

## ICGC MMML-Seq Consortium

**Coordination (C1)** Reiner Siebert[13,35], Susanne Wagner[35], Andrea Haake[35], Julia Richter[35,36] & Gesine Richter[35]

**Data Center (C2)** Roland Eils[4,37,38], Chris Lawerenz[37], Jürgen Eils[37], Jules Kerssemakers[37], Christina Jaeger-Schmidt[37] & Ingrid Scholz[37]

**Clinical Centers (WP1)** Anke K. Bergmann[35,39], Christoph Borst[40], Friederike Braulke[26], Birgit Burkhardt[41,42], Alexander Claviez[39], Martin Dreyling[43], Sonja Eberth[43], Hermann Einsele[44], Norbert Frickhofen[45], Siegfried Haas[40], Martin-Leo Hansmann[46], Dennis Karsch[47], Nicole Klepl[26], Michael Kneba[47], Jasmin Lisfeld[41],

Luisa Mantovani-Löffler[48], Marius Rohde[41], German Ott[49], Christina Stadler[26], Peter Staib[50], Stephan Stilgenbauer[51], Lorenz Trümper[26] & Thorsten Zenz[52]

**Normal Cells (WPN)** Martin-Leo Hansmann[46], Dieter Kube[26], Ralf Küppers [20,21] & Marc Weniger[20,21]

**Pathology and Analyte Preparation (WP2-3)** Siegfried Haas[40], Michael Hummel[27], Wolfram Klapper[36], Ulrike Kostezka[53], Dido Lenze[27], Peter Möller[54], Andreas Rosenwald[55], German Ott[49] & Monika Szczepanowski[36]

**Sequencing and genomics (WP4-7)** Ole Ammerpohl[13,35], Sietse M. Aukema[35,36], Vera Binder[56], Arndt Borkhardt[56], Andrea Haake[35], Jessica I. Hoell[56], Ellen Leich[55], Peter Lichter[57], Cristina López[13,35], Inga Nagel[35], Jordan Pischimariov[55], Bernhard Radlwimmer[57], Julia Richter[35,36], Philip Rosenstiel[58], Andreas Rosenwald[55], Markus Schilhabel[58], Stefan Schreiber[59], Inga Vater[35], Rabea Wagener[13,35] & Reiner Siebert[13,35]

**Bioinformatics (WP8-9)** Stephan H. Bernhart[60,61,62], Hans Binder[60,61], Benedikt Brors[5], Gero Doose[60,61,62], Roland Eils[4,37,38], Steve Hoffmann[60,61,62], Lydia Hopp[60], Daniel Hübschmann[5,8,9,10,11], Kortine Kleinheinz[37,38], Helene Kretzmer[60,61,62], Markus Kreuz[63], Jan Korbel[64], David Langenberger[60,61,62], Markus Loeffler[64], Maciej Rosolowski[64], Matthias Schlesner[18,19], Peter F. Stadler[60,61,62,65,66,67] & Stephanie Sungalee[64]

[35]Institute of Human Genetics, Christian-Albrechts-University, Kiel, Germany. [36]Institute of Pathology, Hematopathology Section, Christian-Albrechts-University, Kiel, Germany. [37]Division of Theoretical Bioinformatics (B080), German Cancer Research Center (DKFZ), Heidelberg, Germany. [38]Department for Bioinformatics and Functional Genomics, Institute of Pharmacy and Molecular Biotechnology and Bioquant, University of Heidelberg, Heidelberg, Germany. [39]Department of Pediatrics, University Hospital Schleswig-Holstein, Campus Kiel, Kiel, Germany. [40]Department of Internal Medicine/Hematology, Friedrich-Ebert-Hospital, Neumünster, Germany. [41]Pediatric Hematology and Oncology, University Hospital Muenster, Muenster, Germany. [42]Pediatric Hematology and Oncology, University Hospital Giessen, Giessen, Germany. [43]Department of Medicine III - Campus Grosshadern, University Hospital Munich, Munich, Germany. [44]Department of Medicine and Poliklinik II, University Hospital Würzburg, Würzburg, Germany. [45]Department of Medicine III, Hematology and Oncology, Dr. Horst-Schmidt-Kliniken of Wiesbaden, Wiesbaden, Germany. [46]Senckenberg Institute of Pathology, University of Frankfurt Medical School, Frankfurt am Main, Germany. [47]Department of Internal Medicine II: Hematology and Oncology, University Medical Center Schleswig-Holstein, Campus Kiel, Kiel, Germany. [48]Hematology and Oncology, Hospital of Internal Medicine II, St-Georg Hospital Leipzig, Leipzig, Germany. [49]Department of Pathology, Robert-Bosch-Hospital, Stuttgart, Germany. [50]Clinic for Hematology and Oncology, St.-Antonius-Hospital, Eschweiler, Germany. [51]Department for Internal Medicine III, University of Ulm and University Hospital of Ulm, Ulm, Germany. [52]National Centre for Tumor Disease, Heidelberg, Germany. [53]Comprehensive Cancer Center Ulm (CCCU), University Hospital Ulm, Ulm, Germany. [54]Institute of Pathology, University of Ulm and University Hospital of Ulm, Ulm, Germany. [55]Institute of Pathology, Comprehensive Cancer Center Mainfranken, University of Würzburg, Würzburg, Germany. [56]Department of Pediatric Oncology, Hematology and Clinical Immunology, Heinrich-Heine-University, Düsseldorf, Germany. [57]Division of Molecular Genetics, German Cancer Research Center (DKFZ), Heidelberg, Germany. [58]Institute of Clinical Molecular Biology, Christian-Albrechts-University, Kiel, Germany. [59]Department of General Internal Medicine, University Kiel, Kiel, Germany. [60]Interdisciplinary Center for Bioinformatics, University of Leipzig, Leipzig, Germany. [61]Department of Computer science, University of Leipzig, Leipzig, Germany. [62]Transcriptome Bioinformatics, LIFE Research Center for Civilization Diseases, University of Leipzig, Leipzig, Germany. [63]Institute for Medical Informatics Statistics and Epidemiology, University of Leipzig, Leipzig, Germany. [64]EMBL Heidelberg, Genome Biology, Heidelberg, Germany. [65]RNomics Group, Fraunhofer Institute for Cell Therapy and Immunology IZI, Leipzig, Germany. [66]Santa Fe Institute, Santa Fe, NM, USA. [67]Max-Planck-Institute for Mathematics in Sciences, Leipzig, Germany.

