## [Peer Review File · Nature Communications]

The genomic and transcriptional landscape of primary central nervous system lymphomaEditorial Note: The item on p14 of this Peer Review File has been redacted as indicated to maintain the confidentiality of unpublished data. The item on p35 has been redacted as indicated to remove third-party material where no permission to publish could be obtained.

REVIEWER COMMENTS

Reviewer #1 (Remarks to the Author): Expert in primary central nervous system lymphomas

The authors have utilized an integrative multiplatform genomic profiling approach to 51 cases of CNS lymphoma including whole genome sequencing, RNA and DNA sequencing, and assessment of copy number alterations. The focus of their research was to compare the most common tumor EBV- PCNSL to other lymphomas with either clinical or biologic similarities including EBV+ CNSL, secondary CNS lymphoma, and ABC DLBCL. The comprehensive nature of their approach is unprecedented in the published literature to date.

The reason that this work is important is that a number of new targeted agents including BTK inhibitors, immunomodulatory agents, and immunotherapy agents (including PD-1 inhibitors and chimeric antigen receptor T-cell therapy) are being developed in clinical trials for patients with lymphomas involving the CNS. The underlying mechanisms of CNS tropism for DLBCL as well as the most critical signaling pathways in PCNSL are understudied. More precise understanding the heterogeneity can lead to precision medicine trials and/or novel therapeutic agents.

Noteworthy findings include:

- 1) Confirmation of the MCD subtype predominately underlying PCNSL
- 2) Novel finding of potential biologic differences in CNS lymphomas involving the meninges compared to brain parenchyma
- 3) Confirmation of distinctive biology of EBV+ CNSL (i.e. lack of MYD88 L265P)
- 4) Both similarities and differences between PCNSL and ABC DLBCL (deletions in 6p21, TERT gene expression, distinct RNA expression signature)
- 5) Description and origin of kataegis events in PCNSL
- 6) Expanded on the the mutational landscape in PCNSL showing higher rates of certain driver mutations
- 7) Novel description of non-coding alterations in RNA

The paper is well-written, well-referenced, clear and concise. I have the following comments designed to improve an already excellent and significant manuscript.

COMMENTS:

P6, LINE 131: I would expand the definition of SCNSL to include synchronous cases (present in both CNS and systemic) and add the important detail that SCNSL include all lymphomas with CNS tropism so it is not a homogenous entity recognized by the WHO. Most are DLBCL.

LINE 135: need a reference for the statement PCNSL in immunocompromised is typically EBV+

LINE 136-137: this sentence is awkward and unclear. Please make more concise and clear because I do think it is an important point for the introduction.

LINE 149: missing a reference PMID: 32187361

LINE 154: missing a reference PMID: 30630772

LINE 159: I would support this statement with original references instead of a review: PMID: 28552327, PMID: 28619981

LINE 172: PCNSL misspelled

LINE 202: similar to above, please make clear that SCNSL is not a WHO classified tumor but a group of entities.

LINE 521: missing reference PMID: 33202420

Reviewer #2 (Remarks to the Author): Expert in lymphoma genomics

Radke, Ishaque and colleagues perform combined genetic and gene expression profiling of 51 lymphomas, primarily 41 EBV-negative PCNSL. The study charts the mutational profile of PCNSL and reaffirms the exonic findings from previous smaller studies. The major advancement in this study is the whole genome approach paired with transcriptomic analyses that, in my knowledge, has not been performed previously and I commend the authors for this together with the suite of analytics required. They have identified interesting and novel differences between PCNSL intraparenchymal, PCNSL-meningeal, systemic DLBCL and SCNSL – although I feel this has to come through more strongly in the manuscript. There are a number of areas that need to be addressed below to maximise the numerous observations noted.

Main comments:

- As might be predicted, the majority of the PCNSLs molecularly type as the MCD type using the LymphGen algorithm. It was notable that more than 20% were not typed – were there any common or unifying genetic features of this group? This should be elaborated on and discussed.
- I was surprised at the high level of ‘tumor in normal’ contamination in this cohort particularly as the germline used in this study were blood samples. Prior studies examining this, even in the context of metastatic tumors (e.g using the deTiN tool) did not identify such contamination in blood. What is the ground truth used here? As these lymphomas would not have circulating disease by definition as PCNSLs are localised to the brain/meninges and even on assumption that this might be ctDNA in the peripheral blood, with the limit of detection/methods applied, one would not be able to detect this. Although the majority of the ‘rescued’ somatic variants were non-exonic, could the authors elaborate on the reasoning for this observation? It would be helpful to include the metrics regarding reads supporting control vs tumor in a format that is clearer to the readership as Suppl Table 2 is a little difficult to parse.
- Given that the majority of the coding mutations have been described in prior studies, the authors should perhaps more clearly highlight and focus in their results section on what the new drivers are from their study rather just descriptively report their findings – e.g OSBPL10 seems to be most mutated gene at 83%. I do not believe these have been reported in other studies – what is it/what does it do/where are the mutations located/is the loci impacted by CNAs also? Etc.
- Mutations in the non-coding genes are interesting – MALAT1 and NEAT – was this confirmed in the validation cohort? Is it unique to PCNSL – how does this compare to systemic DLBCL/FL for which WGS data is available? Do these mutations appear clonal/subclonal? – this should be analysed/reported.
- Similar to the point above regarding recurrent SVs – rather than just reporting the variants, please put into context/compare with systemic DLBCL/SCNSL – this will give an appreciation of where PCNSL sits biologically in the spectrum of these disorders.
- Whilst many of the figures visually are very informative, I found Figure 7 difficult to follow and this needs to be simplified – for example 7A – there is too much going on in the figure – e.g the ‘Age’ row does not appear to add much. It would be better just to present what the disease types are in this row. In fig 7B – what are the headings for these two groups (blue and red??) – the row for subtype 1 and 2 what do these mean as each sample is black or green?
- What is subtype 1 and subtype 2 referring to – is it the 2 PCNSL groups a ‘pure’ vs ‘impure’ as the definition of subtype 1 and 2 are not mentioned in the main text? In the results section, the authors suggest that the ‘impure’ group’s gene expression closely resembles the normal brain CNS controls but this does not appear to be the case in fig 7B if I have interpreted this correctly and is actually the reverse. Please clarify. The gradient for the genomic purity which I presume is tumor purity appears to have samples coloured as white – does this mean zero tumor content? If so, should these samples be included in the analyses at all? How have the authors defined impurity? Furthermore, if impurity is a major issue with a subset of the cohort, shouldn’t these be excluded from the analyses so the focus can be on PCNSL with sufficient tumor content to dissect the molecular groupings more clearly.
- The figure for the telomere content from the supplement should perhaps replace fig 8A as fig 8B is a better depiction of the TERT expression. How did the telomere content correlate with age in this cohort and control samples? Also, the authors correct for tumor purity using a control sample presumably the germline sample (derived from blood). This potentially confounds the analyses as it is now well accepted that telomere length/content varies by tissue type. It is worth assessing if similar results are obtained in an uncorrected analysis. Figure 8D should include the normal GCB

and naïve B cells, and ? normal brain to see how PCNSL compares to these.

Minor comments:

- The results state that the median age at diagnosis is 66, however the supplementary table lists each individual within the study cohort as an 'age range' – I am not sure how the median age was computed based on this. Please include the actual age at diagnosis.
- There is no benefit in stating 'SNVs was significantly higher in PCNSL than in systemic DLBCL' when the median depth in the PCNSL cohort is nearly double that in the DLBCL and confounds the analyses as the authors state themselves. This can be removed from the results section. The metrics for median depths for the different cohorts can just be included in a supplement table. The authors should just state the no of SNVs detected in PCNSL in the results section.
- Although mentioned in the methods section, clinical information regarding treatment and outcomes is not mentioned subsequently and should perhaps be included in the supplement.
- The authors should recheck their p values in fig 8b are correctly stated as they all read "p = 0.027".

Reviewer #3 (Remarks to the Author): Expert in cancer genomics and bioinformatics

Characterizing tumor landscapes using various omic data types is an active area of cancer research and the authors' manuscript is a contribution to this timely milieu for lymphomas of the central nervous system. Here, the authors generated comprehensive genomic data, including WGS on 38 cases, and validated against a large, independent cohort. One then finds the data-centric, informatics-heavy analysis and the extensive cataloging of features that are characteristic of a landscape paper. Overall, this seems to be a descriptive landscape paper.

Among the observations are the highly recurrent somatic alterations in non-coding genes and recurrent copy number alterations, mutational signatures, and new drivers. As with many landscape studies, consequences are set against what is known from the literature, although some of what the investigators report recapitulates known facts, as would be expected. Some aspects, like recurrent SVs are followed-up and some observations have clear translational relevance, like IGL and IGH translocations having nearby breakpoints with CD274 and corresponding observation of heightened PD-L1 expression and the molecular subtyping results. One take-home is that PCNSL and DLBCL are distinguishable in terms of several genomic attributes, including NFkB signaling, which potentially has translational ramifications. Authors need to highlight what the novel findings are.

Questions and Criticisms

1. I'm not sure why the authors use Build 37 throughout, which has been obsolete for some time. It would be best to use the current human reference, or note the specific reasons that necessitate using GRCh37.
2. The equation for correcting telomere content will be difficult to understand to many readers. Please either cite the source from where this was obtained, or show its derivation, for example as can be readily done using an Expected Value approach for tumor and control and the assumption that EV(C) in the tumor is identical with EV(C) in the actual control sample.
3. Line 303: "single base substitution" is mis-spelled
4. The authors seem comfortable with accuracy for their mutation calls having VAF > 10%, but readers will wonder about lower-VAF mutations. This needs to be thoroughly and convincingly addressed.
5. There are quite a few heuristics used in the bioinformatics analysis, for instance lines 258-261. There should be discussions about how some of these were determined and how sensitive the results are to those choices.

Point-by-point responses to the reviewers' comments

We have used different font colours to differentiate between the reviewers' comments (**black font colour**), our responses (**blue font colour**), and revised manuscript passages (**dark orange colour**). We uploaded a manuscript version, where all changes are highlighted in yellow.

Reviewer #1 (Remarks to the Author)

Expert in primary central nervous system lymphomas

The authors have utilized an integrative multiplatform genomic profiling approach to 51 cases of CNS lymphoma including whole genome sequencing, RNA and DNA sequencing, and assessment of copy number alterations. The focus of their research was to compare the most common tumor EBV- PCNSL to other lymphomas with either clinical or biologic similarities including EBV+ CNSL, secondary CNS lymphoma, and ABC DLBCL. The comprehensive nature of their approach is unprecedented in the published literature to date.

The reason that this work is important is that a number of new targeted agents including BTK inhibitors, immunomodulatory agents, and immunotherapy agents (including PD-1 inhibitors and chimeric antigen receptor T-cell therapy) are being developed in clinical trials for patients with lymphomas involving the CNS. The underlying mechanisms of CNS tropism for DLBCL as well as the most critical signaling pathways in PCNSL are understudied. More precise understanding the heterogeneity can lead to precision medicine trials and/or novel therapeutic agents.

Noteworthy findings include:

- 1) Confirmation of the MCD subtype predominately underlying PCNSL
- 2) Novel finding of potential biologic differences in CNS lymphomas involving the meninges compared to brain parenchyma
- 3) Confirmation of distinctive biology of EBV+ CNSL (i.e. lack of MYD88 L265P)
- 4) Both similarities and differences between PCNSL and ABC DLBCL (deletions in 6p21, TERT gene expression, distinct RNA expression signature)
- 5) Description and origin of kataegis events in PCNSL
- 6) Expanded on the the mutational landscape in PCNSL showing higher rates of certain driver mutations
- 7) Novel description of non-coding alterations in RNA

The paper is well-written, well-referenced, clear and concise. I have the following comments designed to improve an already excellent and significant manuscript.

Response: We thank reviewer #1 for his/her overall positive evaluation of our study as well as for the valuable comments that have helped to substantially improve our manuscript.

COMMENTS:

1.1 P6, LINE 131: I would expand the definition of SCNSL to include synchronous cases (present in both CNS and systemic) and add the important detail that SCNSL include all lymphomas with CNS tropism so it is not a homogenous entity recognized by the WHO. Most are DLBCL.

Response: We expanded the definition of SCNSL according to the reviewer's suggestions and cited the following publications:

- 5 Ferreri, A. J. Risk of CNS dissemination in extranodal lymphomas. *The Lancet Oncology* **15**, e159-169, doi:10.1016/S1470-2045(13)70568-0 (2014).
- 6 Malikova, H. *et al.* Secondary central nervous system lymphoma: spectrum of morphological MRI appearances. *Neuropsychiatr Dis Treat* **14**, 733-740, doi:10.2147/NDT.S157959 (2018).

Revised (in Introduction): [...] *presenting initially with systemic, non-CNS or synchronous systemic and CNS involvement. The term SCNSL comprises all systemic lymphomas that spread to the CNS and its presentation, tropism, outcome and therapeutic options differ from PCNSL (3, 4). Typically, SCNSL are classified as diffuse large B-cell lymphoma (DLBCL), while other types such as follicular lymphoma, T-cell lymphoma or Hodgkin lymphoma are extremely rare (5, 6).*

Revised (in Material and Methods section, CNSL study cohort): *In our SCNSL cohort, three patients presented with initial lymph node manifestation, one patient with testicular involvement, and three patients with involvement of parotid gland, or liver, or urinary tract, respectively.*

1.2 LINE 135: need a reference for the statement PCNSL in immunocompromised is typically EBV+

Response: We cited the following publications to support our statement:

- 1 Louis, D. N., Ohgaki, H., Wiestler, O. D. & Cavenee, W. K. WHO classification of tumours of the central nervous system. 4 edn, Vol. 1 (International agency for research on cancer Lyon, France, 2016)
- 7 Bashir, R., McManus, B., Cunningham, C., Weisenburger, D. & Hochberg, F. Detection of Eber-1 RNA in primary brain lymphomas in immunocompetent and immunocompromised patients. *J Neurooncol* **20**, 47-53, doi:10.1007/BF01057960 (1994)
- 8 Kleinschmidt-DeMasters, B. K., Damek, D. M., Lillehei, K. O., Dogan, A. & Giannini, C. Epstein Barr virus-associated primary CNS lymphomas in elderly patients on immunosuppressive medications. *Journal of neuropathology and experimental neurology* **67**, 1103-1111, doi:10.1097/NEN.0b013e31818beaea (2008).

1.3 LINE 136-137: this sentence is awkward and unclear. Please make more concise and clear because I do think it is an important point for the introduction.

Response: We apologize for not having been clear enough in the original version of our manuscript. To improve readability we have modified the sentence.

Original: *The mechanisms leading to the topographical restriction of PCNSL are still matter of scientific debate.*

Revised (in Introduction): *The mechanisms leading to the observed topographical restriction of PCNSL to the CNS have yet not been fully elucidated.*

1.4 LINE 149: missing a reference PMID: 32187361

Response: We would like to thank the reviewer for drawing our attention to this publication, which we have cited in the revised manuscript

- 35 Lacy, S. E. *et al.* Targeted sequencing in DLBCL, molecular subtypes, and outcomes: a Haematological Malignancy Research Network report. *Blood* **135**, 1759-1771, doi:10.1182/blood.2019003535 (2020).

1.5 LINE 154: missing a reference PMID: 30630772

Response: We would like to thank the reviewer for drawing our attention to this publication. We have now cited this paper in the revised manuscript.

- 42 Bromberg, J. E. C. *et al.* Rituximab in patients with primary CNS lymphoma (HOVON 105/ALLG NHL 24): a randomised, open-label, phase 3 intergroup study. *The Lancet. Oncology* **20**, 216-228, doi:10.1016/S1470-2045(18)30747-2 (2019).

1.6 LINE 159: I would support this statement with original references instead of a review: PMID: 28552327, PMID: 28619981

Response: We would like to thank the reviewer for drawing our attention to these publications, which we have cited in the revised manuscript.

- 50 Lionakis, M. S. *et al.* Inhibition of B Cell Receptor Signaling by Ibrutinib in Primary CNS Lymphoma. *Cancer Cell* **31**, 833-843 e835, doi:10.1016/j.ccell.2017.04.012 (2017).
- 51 Grommes, C. *et al.* Ibrutinib Unmasks Critical Role of Bruton Tyrosine Kinase in Primary CNS Lymphoma. *Cancer Discov* **7**, 1018-1029, doi:10.1158/2159-8290.CD-17-0613 (2017).

1.7 LINE 172: PCNSL misspelled

Response: We corrected this.

1.8 LINE 202: similar to above, please make clear that SCNSL is not a WHO classified tumor but a group of entities.

Response: We have changed the sentence accordingly.

Original: *The inclusion criteria were based on the diagnosis of PCNSL and SCNSL according to the recent WHO classifications of tumors of hematopoietic and lymphoid organs and tumors of the central nervous system.*

Revised (in Material and Methods, CNS lymphoma (CNSL) study cohort): We included DLBCL confined to the CNS as PCNSL according to the recent WHO classifications of tumors of hematopoietic and lymphoid organs and tumors of the central nervous system. DLBCL, which presented initially with systemic, non-CNS or synchronous systemic and CNS involvement were included as SCNSL (1,2,11,62,63). In our SCNSL cohort, three patients presented with initial lymph node manifestation, one patient with testicular involvement, and three patients with involvement of parotid gland, or liver, or urinary tract, respectively.

1.9 LINE 521: missing reference PMID: 33202420

Response: We cited the following reference.

- 103 Gandhi, M. K. et al. EBV-associated primary CNS lymphoma occurring after immunosuppression is a distinct immunobiological entity. *Blood* 137, 1468-1477, doi:10.1182/blood.2020008520 (2021)

Reviewer #2 (Remarks to the Author)

Expert in lymphoma genomics

Radke, Ishaque and colleagues perform combined genetic and gene expression profiling of 51 lymphomas, primarily 41 EBV-negative PCNSL. The study charts the mutational profile of PCNSL and reaffirms the exonic findings from previous smaller studies. The major advancement in this study is the whole genome approach paired with transcriptomic analyses that, in my knowledge, has not been performed previously and I commend the authors for this together with the suite of analytics required. They have identified interesting and novel differences between PCNSL intraparenchymal, PCNSL-meningeal, systemic DLBCL and SCNSL – although I feel this has to come through more strongly in the manuscript. There are a number of areas that need to be addressed below to maximise the numerous observations noted.

Response: We thank reviewer #2 for his/her remarks and valuable comments that have helped to substantially improve our manuscript.

Main comments:

2.1 As might be predicted, the majority of the PCNSLs molecularly type as the MCD type using the LymphGen algorithm. It was notable that more than 20% were not typed – were there any common or unifying genetic features of this group? This should be elaborated on and discussed.

Response: We thank reviewer #2 for this comment but we are not entirely sure what “not typed” refers to. We believe that reviewer #2 may refer to those samples that 1) were not typed at all (i.e. classified as “N/A”) or 2) or he may refer to samples that were classified as “Other”. We thus aimed to cover both types of interpretation as follows:

1. This critique obviously is a result of a lack of clarity in the original manuscript text and the original Supplementary Table 1, for which we apologize. We predicted the molecular subtype for all specimens, where DNA and whole-genome sequencing data was available. In some cases (around 20% of all cases), only RNA was available and therefore, only RNA sequencing could be done. These 20% of samples could thus not be typed according to the LymphGen algorithm and, consequently, were listed as “NA” in the revised Supplementary table 1.

To clarify this issue, we have included a statement in the revised manuscript (in Material and Methods, LymphGen algorithm): *Samples, where only RNA was available were listed as “NA” in Supplementary table 1.*

2. We further investigated the PCNSL cases classified by LymphGen as “Other” samples and compared the results to those PCNSL cases classified by LymphGen as “MCD”. The following paragraph was added to the revised manuscript (in Results, PCNSL represent MCD genetic subtype of DLBCLs):

*PCNSL samples classified as “Other” exhibited different CNV profiles affecting chromosome arms 1q, 2p, 2q, 3q, 4p, and 11p, as well as significantly more deletions of CREBBP compared to PCNSL samples classified as MCD by the LymphGen algorithm (**Supplementary figure 2 A**). CREBBP inactivation is considered an early event in FLs and a subset of systemic DLBCL, mostly of GCB origin (91-95). CREBBP inactivation is also described as a hallmark of the EZB class, but LymphGen’s classification model is restricted to CREBBP point mutations and not focal deletions. The finding of a significantly increase number of CREBBP alterations ($p=0.046$, Mann-Whitney U test) in PCNSLs classified as “Other” compared to MCD might, thus, imply a small subset of PCNSL to more resemble GCB-like DLBCL or, alternatively, the existence of a group of occult systemic GCB-lymphomas with first clinical presentation in the CNS. Additionally, PCNSL-Other demonstrated significantly fewer mutations in GRHPR, ETV6, and PIM1 (**Supplementary figures 2 B, C**).*

The following citations were added to the revised manuscript:

- 91 Zhang, J. et al. The CREBBP Acetyltransferase Is a Haploinsufficient Tumor Suppressor in B-cell Lymphoma. *Cancer Discov* 7, 322-337, doi:10.1158/2159-8290.CD-16-1417 (2017).
- 92 Pasqualucci, L. et al. Inactivating mutations of acetyltransferase genes in B-cell lymphoma. *Nature* 471, 189-195, doi:10.1038/nature09730 (2011).
- 93 Meyer, S. N. et al. Unique and Shared Epigenetic Programs of the CREBBP and EP300 Acetyltransferases in Germinal Center B Cells Reveal Targetable Dependencies in Lymphoma. *Immunity* 51, 535-547 e539, doi:10.1016/j.immuni.2019.08.006 (2019).
- 94 Schmidt, J. et al. CREBBP gene mutations are frequently detected in in situ follicular neoplasia. *Blood* 132, 2687-2690, doi:10.1182/blood-2018-03-837039 (2018).
- 95 Loeffler, M. et al. Genomic and epigenomic co-evolution in follicular lymphomas. *Leukemia* 29, 456-463, doi:10.1038/leu.2014.209 (2015).

We added an additional Supplementary figure to the revised manuscript (**now Supplementary figure 2**, please see below):

Revised Supplementary Figure 2: Genomic structural variation in PCNSL-MCD and PCNSL-Other.

Relative prevalence of somatic copy number aberrations in tumour samples (**A**, middle panel), showing presence of at least one copy number gain (orange bars), copy number loss (blue bars), as a proportion of analysed samples. The differences between PCNSL-MCD and PCNSL-Other are highlighted in red and some candidate genes detected to be significant by Gistic2 (q -value < 0.25) are shown. PCNSL-Other demonstrated significantly more deletions in *CREBBP* ($p=0.04648$, One-tailed Mann-Whitney U test) compared to PCNSL-MCD. The dot plots (**B**, **C**) show the log₂ fold change (colour) and significance (size of

dot) of alteration frequencies of genes in PCNSL compared to different subcohorts, PCNSL-MCD, and PCNSL-Other.

2.2 I was surprised at the high level of 'tumor in normal' contamination in this cohort particularly as the germline used in this study were blood samples. Prior studies examining this, even in the context of metastatic tumors (e.g using the deTiN tool) did not identify such contamination in blood. What is the ground truth used here? As these lymphomas would not have circulating disease by definition as PCNSLs are localised to the brain/meninges and even on assumption that this might be ctDNA in the peripheral blood, with the limit of detection/methods applied, one would not be able to detect this. Although the majority of the 'rescued' somatic variants were non-exonic, could the authors elaborate on the reasoning for this observation? It would be helpful to include the metrics regarding reads supporting control vs tumor in a format that is clearer to the readership as Suppl Table 2 is a little difficult to parse.

Response: We thank the reviewer for bringing up this undeniably very interesting point, especially given the perturbed blood-brain barrier in PCNSL patients.

We found on average 31 somatic SNVs to be "rescued" in PCNSL cases, using the TiNDA algorithm for tumor in normal analysis. This number constitutes only 0.14% of the total mutations found (from an average of 22,198 SNVs identified in PCNSL samples). We believe that this level reflects a very low level of tumor in normal, which is similar to observations in various tumor entities analyzed via deTiN in the ICGC PCAWG study (<https://www.nature.com/articles/s41586-020-1969-6/figures/3>). Therefore, we do have good reasons to propose that there were no substantial tumor in normal contaminations in the specimens enclosed within our study, and that the deTiN tool indeed was capable to find such contaminations in the blood controls for other tumor entities.

The question regarding the ground truth is certainly an interesting one. We were also very excited to see some evidence for clonal tumor mutations at low VAF in the blood control samples, as this may have implicated the presence of tumor cells in the peripheral blood. However, we have seen very similar levels of tumor in normal in nearly all other comparable studies performed at the DKFZ and Charité (including several ICGC projects). This could either imply that (i) there are indeed many circulating tumor cells in the peripheral blood for a diverse range of tumor entities, or (ii) the general prevalence of detecting tumor in normal is a sample handling artifact or (iii) occurrence of lymphoma-like mutations in non-neoplastic (immune) cells. For PCNSL, this is particularly relevant due to the perturbed blood brain barrier (BBB). In fact, surgery-induced BBB disturbance may facilitate tumor DNA leakage into the circulation, which may lead to detection of *MYD88* and *CD79B* mutations on cfDNA (PMID: 32745612). Moreover, lymphoma driver mutations might also be present in non-neoplastic diseases (PMID: 32059783).

To demonstrate that we see evidence for tumor in normal blood in other malignant primary brain tumor entities where extracranial metastases are rare, we analyzed other publicly available datasets (from e.g. gliomas and pediatric medulloblastoma (**Point-by-point response Figures 1 and 2**, please see below)).

Point-by-point response Figure 1: Example of the SNV VAF modeling using the TiNDA algorithm for a PCNSL sample. Each SNV is modelled based on Tumor and Control VAF. Clustering analysis reveals a population of SNVs that have a higher VAF in tumor (blue dots), against those that have higher or expected VAF in the control (pink dots). The TiNDA algorithm also requires input of likely somatic mutations in order to correctly model the VAF for “rescuable mutations”, resulting in more observable blue dots than rescued mutations.

[REDACTED]

Each dataset demonstrates that non-negligible amounts of tumor signal are found in controls. Since we were unable to discern the true cause of medium/low-level presence of clonal tumor mutations in the control, thus being uncertain to undeniably assign a true biological meaning to these alterations, we decided to include this approach exclusively as a methodology to rescue mutations.

Moreover, the majority of mutations rescued via the TiNDA algorithm are non-exonic as this represents the expected given the somatic SNV distribution in the samples. To clarify this in the text we have added the sentence in the revised manuscript (in Material and Methods, Tumor in normal contamination detection):

In our series, we only found a very low level of tumor in normal. The rescued mutations followed a genomic distribution similar to the overall mutational landscape of PCNSL. We observed that rescued mutations were 1% exonic (compared to 1% in the mutational landscape), 32% intronic (c.f. 33%), 53% intergenic (c.f. 55%), 13% on ncRNA (c.f. 12%).”

We agree that adding the control and tumor VAF for rescued mutations will help to appreciate these data more easily. We have therefore added additional columns to the revised **Supplementary table 2** to depict control VAF and tumor VAF, where the median control VAF for rescued mutations per patient sample is 0.057 (range 0.033-0.11).

2.3 Given that the majority of the coding mutations have been described in prior studies, the authors should perhaps more clearly highlight and focus in their results section on what the new drivers are from their study rather just descriptively report their findings – e.g OSBPL10 seems to be most mutated gene at 83%. I do not believe these have been reported in other studies – what is it/what does it do/where are the mutations located/is the loci impacted by CNAs also?

Response: We appreciate the reviewer’s comment to emphasize more prominently novel drivers in the results section. Indeed, our original manuscript comprises only 1 paragraph on known PCNSL/MCD drivers, which is followed by describing (i) enrichment of genes regulated by *BCL6* in the driver genes, (ii) novel co-occurrence patterns of *TBL1XR1* with *MYD88*, followed by reporting on *TBL1XR1* as having a role in modulating TLR/MYD88 signaling, (iii) novel mutual exclusivity patterns of *MYD88* mutations with *SPEN*, and (iv) novel drivers in our PCNSL series. We believe that these findings justify the reasoning of our study and the importance of our data.

Going beyond the known, we identified a total of 50 mutated driver genes, of which only 21 were known and previously reported drivers. These drivers are depicted in an oncoprint plot (**Main Figure 2 D**), which also contextualizes the genomic lesions in terms of structural and copy number

aberrations. We provide evidence for additional mutated driver genes of which some have only been reported in the context of aberrant SHM (aSHM) in PCNSL (e.g. *PIM1*, *OSBPL10*; PMID: 25189415). We also systematically compared our drivers to those identified for the MCD type DLBCL as reported by Wright et al. In doing so, we found genes that were not reported by Wright et al (*FBXW7*, *ATM*, *TMSB4X*, *THRAP3*, *ID2*, *GRB2*, *ZEB2*, *GLI3*, *UBA1*, *MAPKAPK2*, *AXIN2*, *TAP2*, *ROCK1*, *CEP290*, and *HLA-DQB1*), as well as genes that were previously reported, but detected at different frequencies in our cohort (*PABPC1* (10% vs 0%), *P2RY8* (13% vs 1.2%), *ITPKB* (23% vs 2.5%), *GNA13* (20% vs 5.1%), and *B2M* (13.3% vs 2.8%)). In addition, we also compared drivers identified in our series to those of other non-MCD classes identified by Wright et al: BN2 (*CCND3*, *BCL6*, *HIST1H1D*, *SPEN*, *PABPC1*, and *UBE2A*), EZB (*GNA13*, *IRF8*, *BCL7A*, *KM2TD*, *EP300*), ST2 (*P2RY8*, *TET2*, *ZFP36L1*, and *ITPKB*) and A53 (*B2M* and *TP53*).

We also specifically assessed mutations in *OSBPL10* as suggested by this reviewer. *OSBPL10* has been shown to be mutated in PCNSL (PMID: 25189415) and DLBCL (PMID: 29731965, 29641966, 23292937, 26608593, 26647218), especially in the MCD/C5 subtype (PMID: 33657296). However, *OSBPL10* mutations have previously been linked to aSHM in PCNSL (PMID: 25189415). Most of the identified mutations in DLBCL were confined to the exon 1 coding region (PMID: 29731965), consistent with our observations. Furthermore, we did not find *OSBPL10* to be enriched for copy number alterations. To clarify this in the manuscript text, we have revised the text by clearly stating that *OSBPL10* has been described previously as follows (in Results, Driver mutations in CNSL):

OSBPL10 was previously reported as a target of aSHM in PCNSL (53). Consistent with observations in DLBCL (102), most of the identified mutations in PCNSL were confined to the coding region in exon 1 (Supplementary Figure 3).

The following citations were added:

- 53 Vater, I. et al. The mutational pattern of primary lymphoma of the central nervous system determined by whole-exome sequencing. *Leukemia* 29, 677-685, doi:10.1038/leu.2014.264 (2015).
- 102 Dobashi, A. et al. TP53 and OSBPL10 alterations in diffuse large B-cell lymphoma: prognostic markers identified via exome analysis of cases with extreme prognosis. *Oncotarget* 9, 19555-19568, doi:10.18632/oncotarget.24656 (2018)

We added an additional Supplementary figure (now **Supplementary Figure 3**) to display the *OSBPL10* mutations on our cohort:

Revised Supplementary Figure 3: Somatic SNV and CNA landscape around *OSBPL10*. Somatic single nucleotide variants (SNVs) from samples in our study indicate enrichment of C>T mutations in the first exon of the *OSBPL10* gene, consistent with aberrant somatic hypermutation (aSHM) in previous studies. For the SNV panel, each somatic mutation observed in a sample is marked, and at this resolution, the clustered mutations in the *OSBPL10* promoter appear as a block. In-depth breakdown of mutational frequencies between stratified subcohorts are shown in dot plots in the manuscript (Figure 2 B). The copy number alterations (CNAs) show that very few samples are affected by copy number changes irrespective of cohort stratification. The CNA panels are normalized to the stratified sub-cohort size, with red indicating cumulative amplifications in samples, and blue indicating cumulative deletions.

To further substantiate this finding, we performed bidirectional Sanger sequencing aimed at validating the WGS results. We focused on mutations in the region of chromosome 3, positions 32022426-32022670, and we validated 87% of mutations (52 of 60). Of the mutations that could not be validated, some occurred within a 24 bp hotspot in between chromosome 3, positions 32022589-32022613, which may be due to this region overlapping a (CTG)_n repeat element. By excluding this region the validation improves to 91% (39 of 43).

We added the results to a revised Supplementary table 4.

We agree with the reviewer that it is indeed important to highlight novel findings, and as such we reviewed our data and believe that we previously undersold the potential importance of MYC as a driver in PCNSL. We have revised the manuscript text accordingly:

*A remarkable finding was the high frequency of MYC mutations in PCNSL in the absence of MYC translocations. MYC alteration does not belong to the defining feature of the LymphGen algorithm nor has it been described as a driver in DLBCL by Chapuy et al. (29), though its functional relevance as oncogene in DLBCL has been shown by Reddy et al. (106). Mutation of MYC in lymphomas is frequently linked to IGH translocations, which nevertheless are rare in the PCNSL as shown in the present as well as previous studies (23,107). Whereas previous studies showing a high frequency of MYC mutations in PCNSL focused on the region underlying SHM in PCNSL (108), we here show that these mutations scatter across the gene (**Supplementary figure 4**). The function of the changes remains elusive but it is intriguing to speculate that at least part of them might contribute to the "double expression" of BCL2 and MYC in the absence of MYC translocation in PCNSL which has been associated with unfavorable outcome in systemic DLBCL (109).*

The following citations were added to the revised manuscript:

- 107 Nosrati, A. et al. MYC, BCL2, and BCL6 rearrangements in primary central nervous system lymphoma of large B cell type. *Annals of hematology* 98, 169-173, doi:10.1007/s00277-018-3498-z (2019)
- 108 Montesinos-Rongen, M., Van Roost, D., Schaller, C., Wiestler, O. D. & Deckert, M. Primary diffuse large B-cell lymphomas of the central nervous system are targeted by aberrant somatic hypermutation. *Blood* 103, 1869-1875, doi:10.1182/blood-2003-05-1465 (2004)
- 109 Brunn, A. et al. Frequent triple-hit expression of MYC, BCL2, and BCL6 in primary lymphoma of the central nervous system and absence of a favorable MYC(low)BCL2 (low) subgroup may underlie the inferior prognosis as compared to systemic diffuse large B cell lymphomas. *Acta neuropathologica* 126, 603-605, doi:10.1007/s00401-013-1169-7 (2013).

The following additional Supplementary figure (now **Supplementary Figure 4**) was added to the revised manuscript:

Supplementary Figure 4: (A) Mutational analysis on MYC gene and protein. Raw RNAseq reads aligning across the MYC transcript in two PCNSL (LS-0004, LS-0101) and one ABC-DLBCL (4135099, upper panel). The gene model for the canonical transcript for MYC, ENST00000377970.2, marking alternative start sites that encode for the proteins P01106-1 and P01106-2. Distribution of somatic SNVs and indels in MYC (lower panel). Mutations which affect the RGYW/DGYW motifs, are indicated by blue and red dots; grey dots show mutations outside the motifs. Somatic SNVs and indels identified in PCNSL and SCNSL samples which may cause potential protein coding changes on either the canonical or other transcripts of MYC. **(B)** The lollipop plot shows the protein domains (coloured boxes) over protein coding positions of the MYC protein isoform P01106-2 encoded by MycST00000377970.2 (grey bar). The somatic non-silent point mutations observed in the PCNSL samples that mapped to the MYC isoform are shown as red dots, with the protein coding changes annotated above the dot. Each somatic mutation was only observed once in the series. The post translational modification sites from PhosphoSitePlus v6.6.0.2 are shown as other coloured dots. The height of the dots does not encode for recurrence. **(C)** The violin plots show the log10 TPM RNA expression of MYC in the different subgroups.

2.4 Mutations in the non-coding genes are interesting – MALAT1 and NEAT – was this confirmed in the validation cohort? Is it unique to PCNSL – how does this compare to systemic DLBCL/FL for which WGS data is available? Do these mutations appear clonal/subclonal? – this should be analysed/reported.

Response: We thank the reviewer for this remark. Both MALAT1 and NEAT1 have not been linked to PCNSL before but have been implicated in DLBCL. Our results show that the rate of alterations in MALAT1 and NEAT1 is comparable between PCNSL, SCNSL, and DLBCL. The results are displayed in Figure 3, where the oncoprint plot shows the mutations observed in or PCNSL samples, and the corresponding dot plot reflects the log2 fold change and significance of alteration

frequencies in the other subcohorts and RNAseq subgroups compared to PCNSL. The frequency of mutations in ncRNAs is comparable between PCNSL and ABC-type DLBCLs apart from AL122127.1 and AL122127.4 that are both located in the IGH locus, and RP11-211G3.2 which is situated in the first intron of BCL6. To make this clearer in the revised manuscript, we added the following text and three citations to support our statement:

The landscape of mutations affecting ncRNA in PCNSL was comparable to ABC-DLBCL, apart from significantly more mutations in AL122127.1 and AL122127.4 (Figure 3 A), situated in the IGH locus, and in RP11-211G3.2, situated in the first intron of BCL6. While the implications of these mutations are unclear, it is possible that these mutations are accumulated as part of the SHM/aSHM process affecting IGH and BCL6.

Both, MALAT1 and NEAT1, which have not been linked to PCNSL before are known to be mutated and highly expressed in DLBCL (34) and predict poor prognosis (113,114).

The following citations were added to support this statement:

- 34 Hubschmann, D. *et al.* Mutational mechanisms shaping the coding and noncoding genome of germinal center derived B-cell lymphomas. *Leukemia*, doi:10.1038/s41375-021-01251-z (2021)
- 113 Deng, L. *et al.* Aberrant NEAT1_1 expression may be a predictive marker of poor prognosis in diffuse large B cell lymphoma. *Cancer Biomark* 23, 157-164, doi:10.3233/CBM-160221 (2018)
- 114 Wang, Q. M., Lian, G. Y., Song, Y., Huang, Y. F. & Gong, Y. LncRNA MALAT1 promotes tumorigenesis and immune escape of diffuse large B cell lymphoma by sponging miR-195. *Life Sci* 231, 116335, doi:10.1016/j.lfs.2019.03.040 (2019)

Investigating the subclonality of these strongly implicated ncRNA is indeed interesting. Overall, there is no strong evidence to suggest a role for subclonal *MALAT1* and *NEAT1* mutations in PCNSL. In our series, *MALAT1* exonic mutations appear to be clonal, with an average purity corrected VAF of mutations being 0.5 (lowest per sample was 0.4) and *NEAT1* exonic mutations also appear to be clonal, with an average purity corrected VAF of mutations being 0.5. All but 2 samples (LS-107 and LS-004, 0.33 for both) had an average corrected VAF of above 0.4.

2.5 Similar to the point above regarding recurrent SVs – rather than just reporting the variants, please put into context/compare with systemic DLBCL/SCNSL – this will give an appreciation of where PCNSL sits biologically in the spectrum of these disorders.

Response: We appreciate the reviewer's request to compare PCNSL to DLBCL regarding the SVs and CNVs. Therefore, we revised **Main Figure 4** (please see below) and added a CNV plot for GCB-DLBCL in addition to ABC-DLBCL. Additionally, we revised the manuscript text now comparing the direct SVs which affected the different subtypes (in Results, Recurrent structural variations (SVs)).

*Immunoglobulin gene rearrangements were found in all PCNSL, ABC-DLBCL, and GCB-DBCL cases and affected the IGH (100%, 100%, 100%), IGL (73%, 46%, 31%) and IGK (87%, 54%, 63%) loci. Furthermore, direct SVs affected FHIT (73%, 23%, 38%), CDKN2A (67%, 38%, 25%), BCL6 (37%, 21%, 19%), OSBPL10 (33%, 8%, 13%), ETV6 (33%, 15%, 6%), PAX5 (27%, 0%, 13%), PIM1 (23%, 0%, 6%), TOX (23%, 8%, 19%), BTG2 (23%, 8%, 0%), WWOX (23%, 8%, 25%), as well as CD58 (20%, 8%, 19%; **Figure 4 D, Supplementary table 11**).*

Also, **Supplementary table 11** lists all direct SVs in PCNSL and shows a direct comparison to SCNSL, EBV+ PCNSL, PCNSL-M, ABC-DLBCL, GCB-DLBCL, FL as well as the LymphGen and RNAseq subgroups. In summary, the SV landscape of PCNSL most closely resembles the SV landscape of the DLBCL MCD subtype. We did not compare the CV landscape to SCNSLs though, as the number of samples is too low to permit drawing robust conclusions.

Additionally, we added the following statements to the revised manuscript (in Discussion):

The molecular profile of SCNSL on the other hand corresponds to that of systemic DLBCL.

Main Figure 4 (revised)

2.6 Whilst many of the figures visually are very informative, I found Figure 7 difficult to follow and this needs to be simplified – for example 7A – there is too much going on in the figure – e.g the ‘Age’ row does not appear to add much. It would be better just to present what the disease types are in this row. In fig 7B – what are the headings for these two groups (blue and red??) – the row for subtype 1 and 2 what do these mean as each sample is black or green?

Response: We thank the reviewer for this remark and we agree that the figure and its legend were too busy. We revised **Main Figures 7 A-C** and **Supplementary Figure 6 A (now Supplementary figure 9)** according to the reviewer’s suggestions. “Subtype 1” and “age” have been deleted in the revised figures. We now only show “Subtype 2” (now renamed to “Subtype”, see response 2.7 below) as it reflects the disease types in more detail (e.g. FL (1/2) instead of FL only).

In **Main Figure 7 B**, the “Subtype” reflects either PCNSL (light green) or SCNSL (dark green). As suggested by the reviewer, we added the headings “PCNSL subcluster 1” and “PCNSL subcluster 2” to the two expression groups. Additionally, we changed the colours from red and blue to dark and light grey, respectively, to be in line with the colour scheme from **Main Figure 7 A**, where PCNSL are highlighted in grey. As described in the manuscript text, the first PCNSL expression group (PCNSL subcluster 1) consisted of samples with high tumor cell content (determined by whole-genome sequencing and histopathological analysis). The second PCNSL expression group (PCNSL subcluster 2) contained mainly samples with lower tumor cell content, and expression of its signature gene set was indeed similar to normal brain tissue expression.

2.7 What is subtype 1 and subtype 2 referring to – is it the 2 PCNSL groups a ‘pure’ vs ‘impure’ as the definition of subtype 1 and 2 are not mentioned in the main text? In the results section, the authors suggest that the ‘impure’ group’s gene expression closely resembles the normal brain CNS controls but this does not appear to be the case in fig 7B if I have interpreted this correctly and is actually the reverse. Please clarify. The gradient for the genomic purity which I presume is tumor purity appears to have samples coloured as white – does this mean zero tumor content? If so, should these samples be included in the analyses at all? How have the authors defined impurity? Furthermore, if impurity is a major issue with a subset of the cohort, shouldn’t these be excluded from the analyses so the focus can be on PCNSL with sufficient tumor content to dissect the molecular groupings more clearly.

Response: We thank the reviewer for raising a number of relevant questions which we want to answer point by point.

2.7.1 What is subtype 1 and subtype 2 referring to?

Response: We apologize if this was not clear enough in the original manuscript. Subtype 1 and Subtype 2 were introduced in Supplementary table 1. Subtype 1 referred to the histology (e.g. DLBCL, FL) and Subtype 2 implemented clinical data such as location or systemic manifestation (e.g. PCNSL, SCNSL). To make this clearer, we changed the terms in Supplementary table 1 and the legends of the corresponding figures to “**Histology**” and “**Subtype**” in the revised manuscript (please see **Main Figure 7 (revised)** below).

2.7.2 Is it the 2 PCNSL groups a ‘pure’ vs ‘impure’ as the definition of subtype 1 and 2 are not mentioned in the main text?

Response: In addition to histological or clinical disease types, we can demonstrate that all PCNSL samples form a separate transcriptomics cluster which is distinct from systemic DLBCL and FL (**Main Figure 7 A**). Using an additional round of consensus clustering with normal CNS tissue as control, we identified two PCNSL RNA sequencing groups, namely: “**PCNSL subcluster 1**” and “**PCNSL subcluster 2**”. The purpose of this analysis was to remove gene expression signal from background brain tissue (e.g. strongest in “**PCNSL subcluster 2**”) from true PCNSL gene expression signal (e.g. strongest in “**PCNSL subcluster 1**”). Both, **Main Figure 7 B** and **Supplementary Figure 6 A (now Supplementary figure 9 A)** show these two RNA sequencing groups (PCNSL subcluster 1 and PCNSL subcluster 2) but the consensus clustering method was different. The results for skmeans are given in **Main Figure 7 B**. Skmeans (spherical k-means) is similar to kmeans, but uses "cosine similarity", thus, it should be more biologically relevant for clustering biological data. **Supplementary Figure 6 A (now Supplementary figure 9 A)** shows the results using cola with "ATC" as top-value method (please see <https://academic.oup.com/view-large/figure/228361467/gkaa1146fig2.jpg>).

To be more precise, we deleted the terms “pure” and “impure” in the context of the RNA sequencing groups PCNSL subcluster 1 and PCNSL subcluster 2 as this might be misleading since we determined the tumor cell content (TCC) rather by whole genome sequencing.

Additionally, we added the following headings to the heatmaps: “FL”, “PCNSL”, “DLBCL-ABC” and “DLBCL-GCB” to **Main Figure 7 A** (please see below) and “PCNSL subcluster 1” and “PCNSL subcluster 2” to **Main Figure 7 B** (please see below) and **Supplementary Figure 6 A (now Supplementary figure 9 A)**. We hope that this improves readability.

Main Figure 7 A, B (revised)

2.7.3 The gradient for the genomic purity which I presume is tumor purity appears to have samples coloured as white – does this mean zero tumor content?

Response: Concerning the color gradient for the genomic purity, white means that the tumor cell content is 22%, which is the lowest observed TCC in our PCNSL samples. The TCC results are reported in a revised Supplementary table 6.

2.7.4 If so, should these samples be included in the analyses at all?

Response: We only have three samples with TCC below 30%. The lowest TCC in our cohort is 22%. While a tumor purity of 22% may at first appear to be low, one should consider that these samples have particularly high coverage to improve sensitivity. The coverage for the three samples with TCC below 30% was 68.02x, 76.16x, and 78.57x. Additionally, we were able to detect IGH rearrangements in all PCNSL samples and we detected *CD79B* and *MYD88* mutations in the samples with 22% TCC. Therefore, we believe these are reasonable justification to keep these (few) samples with relatively low tumor purity.

2.7.5 How have the authors defined impurity?

Response: We apologise for the confusion here by switching between TCC and purity in our text. We define purity through tumor cell content (TCC) using ACEseq, which is first introduced in the methods section. The algorithm for calculating TCC is described in the ACEseq manuscript

(<https://www.biorxiv.org/content/10.1101/210807v1.full>). ACESeq was one of the CNA callers used in the ICGC-PCAWG study, and demonstrated good performance in accurately predicting TCC. To prevent further confusion, we have replaced all references to “purity” with tumor cell content (TCC) in the revised manuscript.

2.7.6 Furthermore, if impurity is a major issue with a subset of the cohort, shouldn't these be excluded from the analyses so the focus can be on PCNSL with sufficient tumor content to dissect the molecular groupings more clearly.

We entirely agree with the reviewer. Our approach here was to first identify the PCNSL specific gene expression signature (**Figure 7 A**), and then refine this by removing gene expression signals that result from the background brain tissue (**Figure 7 B**). We found it important to report the results in this way, as to demonstrate that we efficiently control for the background contamination. To make this clearer we have rewritten this section (in Results, PCNSL RNA expression signatures are distinct from systemic DLBCL):

Revised (in Results, PCNSL RNA expression signatures are distinct from systemic DLBCL): *To further exclude an impact of potentially contaminating surrounding CNS tissue on gene expression signatures, we analyzed total RNA from normal brain controls (n = 2) and compared this to PCNSL. To investigate the gradient of various tumor cell contents of samples, we spiked increasing concentrations of RNA from non-diseased brain tissue into a PCNSL sample with very high tumor cell content (0%, 20%, 40%, 60%, 80%). Then, we further stratified the PCNSL group by another round of consensus clustering using two different classification methods (**Figure 7 B**, **Supplementary figures 9 A-C**), which both revealed two groups. The first PCNSL expression group (PCNSL subcluster 1) consisted of samples with high tumor cell content (determined by whole-genome sequencing and histopathological analysis). Expression of its signature gene set did not show similarity to normal brain tissue expression. However, the second PCNSL expression group (PCNSL subcluster 2) contained mainly samples with lower tumor cell content, and expression of its signature gene set was indeed similar to normal brain tissue expression (**Figure 7 B**).*

2.8 The figure for the telomere content from the supplement should perhaps replace fig 8A as fig 8B is a better depiction of the TERT expression. How did the telomere content correlate with age in this cohort and control samples? Also, the authors correct for tumor purity using a control sample presumably the germline sample (derived from blood). This potentially confounds the analyses as it is now well accepted that telomere length/content varies by tissue type. It is worth assessing if

similar results are obtained in an uncorrected analysis. Figure 8D should include the normal GCB and naïve B cells, and ? normal brain to see how PCNSL compares to these.

Response: We agree with the reviewer and revised **Main Figure 8 A** accordingly. As suggested, we replaced **Supplementary figure 7 A (now Supplementary figure 10 A)** with **Main Figure 8 A**. Additionally, we added a boxplot of the telomere content per subcohort to **Main Figure 8** (now **Main figure 8 B**, see below):

Main figure 8 B (revised)

Next, we looked at the correlation between age and telomere content in the control samples. As expected, there is an inverse correlation between age and the telomere content in the control samples (Spearman correlation: -0.32, Pearson correlation: -0.27). We added the corresponding scatterplot to **Supplementary figure 7 (now Supplementary figure 10 D)** please see below):

Supplementary Figure 10 D (revised)

We fully understand the reviewer's concern on using blood as a control as it may confound the telomere content analysis. In an ideal - but only theoretically achievable - experimental set up, correcting for the telomere content would be done using non-diseased tissue of the identical individual. Such a correction would require normal samples that models the B-cell origin of the tumors as well as the brain tissue as adjacent normal. However, we do not have such specimens and for obvious reasons cannot get hold of control tissue e.g. from non-diseased parts of the brain of the same patient (and certainly would not get approval from the ethics committee to retrieve such tissue). Therefore, from our experience, using blood controls at least for inter-patient variability (see also correlation to age) and is supposed to be better than using no control at all.

In **Supplementary Figure 7 A**, we already show the telomere content of just the tumor sample, not corrected by blood control. We integrated the corresponding telomere content boxplots **uncorrected** for the tumor sample in **Supplementary Figure 7 C (now Supplementary figure 10 C**, please see below). As with the T/C log2 ratio, all comparisons were again not significant, so yes, similar results are obtained with an **uncorrected** analysis:

Supplementary Figure 10 C (revised)

As suggested by the reviewer, we revised **Main Figure 8 D** (please see below). First, we now show the TCC-corrected telomere content. Second, we added normal GC B- and naive B-cells (no control correction). Spearman was used to correct for outlier samples and to not have discrepancies due to the log transformation. The results show that *TERT* expression also correlates with telomere content in GC B- and naive B-cells. However, these data did not reach statistical significance, most likely due to low sample size:

Main figure 8 D (revised)

Additionally, we show the telomere content T/C log2 ratio upon TCC correction in **Supplementary Figure 7 F (now Supplementary figure 10 F, see below)**. GC B- and naive B-cells were excluded in this analysis, as blood is not the adequate control for these cell populations.

Supplementary Figure 10 F (revised)

We were able to use non-B-cell controls for the GC B-, and naive B-cells, which were FACS-sorted CD3+CD20- cells from the identical sample, i.e. bulk T-cells (**Point-by-point response Figure 4, see below**). However, given that there is a reasonable likelihood that the various types of controls may result in some degree of confusion by the reader, we decided to rather refrain from presenting all of these controls and leave it up to the reviewers and editor to decide whether or not to implement these data.

Point-by-point response Figure 4

Minor comments:

2.9 The results state that the median age at diagnosis is 66, however the supplementary table lists each individual within the study cohort as an ‘age range’ – I am not sure how the median age was computed based on this. Please include the actual age at diagnosis.

Response: We thank the review for bringing this to our attention. We replaced the age range with the actual age of the patients at the time point of diagnosis in a revised Supplementary table 1. To be more precise, we corrected the information in the manuscript, now stating both the median (69 years) and mean age (66.5 years) at the time point of diagnosis.

2.10 There is no benefit in stating ‘SNVs was significantly higher in PCNSL than in systemic DLBCL’ when the median depth in the PCNSL cohort is nearly double that in the DLBCL and confounds the analyses as the authors state themselves. This can be removed from the results section. The metrics for median depths for the different cohorts can just be included in a supplement table. The authors should just state the no of SNVs detected in PCNSL in the results section.

Response: We fully agree with the reviewer and deleted the following sentence in the revised manuscript (Results, Mutational landscape of central nervous system lymphoma (CNSL)):

The number of detected SNVs was significantly higher in PCNSL compared to systemic DLBCL ($p = 0.018$; median SNVs: 18434 (range: 2,771-98,890)) or FL ($p = 1.2 \times 10^{-11}$; median SNVs: 6049 range: 2139-19751), though different read-depths confound this comparison (median depth DLBCL

(tumor): 32 (range: 27-38), (germline controls): 31 (range: 24-37); median depth FL (tumor): 32 (range: 27-36), (germline controls): 31 (range: 24-45))

As suggested by the reviewer, we included the information on the median depth in the revised Supplementary table (**Supplementary table 6**).

While revising this section, we noticed that some of the results given reflect the mean but not the median. We corrected the numbers and the text accordingly. All metrics analysis are given in **Supplementary table 6**.

2.11 Although mentioned in the methods section, clinical information regarding treatment and outcomes is not mentioned subsequently and should perhaps be included in the supplement.

Response: We integrated clinical information if available. As a detailed clinical follow-up was not always available, we deleted this notion from the Material and Method section. Nevertheless, we included information whether the respective patient had received postoperative adjuvant chemotherapy and provided the clinical information that were available (**Supplementary table 1**). Concerning overall survival, we show a Kaplan-Meier analysis of our cohort in **Figure 1 E**.

Concerning our SCNSL samples, we added the location of the initial peripheral manifestation in the revised manuscript (in Material and Methods CNS lymphoma (CNSL) study cohort):

In our SCNSL cohort, three patients presented with initial lymph node manifestation, one patient with testicular involvement, and three patients with involvement of parotid gland, or liver, or urinary tract, respectively.

2.12 The authors should recheck their p values in fig 8b are correctly stated as they all read “p = 0.027”.

Response: We fully understand the reviewer's concern. Nevertheless, we checked this again and this is indeed correct. The p-values were calculated with a Wilcoxon test and corrected with the Holm method. With a small sample size in non-parametric test, it is - as in our case - possible to observe the same p-values for different comparisons.

Reviewer #3 (Remarks to the Author)

Expert in cancer genomics and bioinformatics

Characterizing tumor landscapes using various omic data types is an active area of cancer research and the authors' manuscript is a contribution to this timely milieu for lymphomas of the central nervous system. Here, the authors generated comprehensive genomic data, including WGS on 38 cases, and validated against a large, independent cohort. One then finds the data-centric, informatics-heavy analysis and the extensive cataloging of features that are characteristic of a landscape paper. Overall, this seems to be a descriptive landscape paper.

Among the observations are the highly recurrent somatic alterations in non-coding genes and recurrent copy number alterations, mutational signatures, and new drivers. As with many landscape studies, consequences are set against what is known from the literature, although some of what the investigators report recapitulates known facts, as would be expected. Some aspects, like recurrent SVs are followed-up and some observations have clear translational relevance, like IGL and IGH translocations having nearby breakpoints with CD274 and corresponding observation of heightened PD-L1 expression and the molecular subtyping results. One take-home is that PCNSL and DLBCL are distinguishable in terms of several genomic attributes, including NFkB signaling, which potentially has translational ramifications. Authors need to highlight what the novel findings are.

Response: We thank the reviewer for acknowledging the translation potential of our study, and for constructive suggestions to our work.

Questions and Criticisms

3.1 I'm not sure why the authors use Build 37 throughout, which has been obsolete for some time. It would be best to use the current human reference, or note the specific reasons that necessitate using GRCh37.

Response: We fully understand the reviewer's point concerning the use of GRCh37. However, we had decided for using this genome build so that results are easy to compare and contextualize as the majority of cancer genomics literature is still based on GRCh37. The ICGC PCAWG study, for example, was published in *Nature* in 2020 and was based on GRCh37. Furthermore, the few big whole genome sequencing studies in lymphomas (e.g. PMIDs: 30275490, 23699601, 23292937, 24145436, 33953289, 30926794) are to the best of our knowledge also based on GRCh37 genome assembly. And while we have the largest PCNSL cohort published to date, compared to

the collective work of international leaders in peripheral blood lymphoma genomics, our study cohort is still rather small. Therefore, increasing comparability to existing data was a very important consideration.

However, the underlying point of the reviewer is still valid and requires attention – does changing the genome assembly to a newer version indeed change results, and how does this change the comparability of our presented results to general literature? We hope to address this in future projects. As for now, the advantage of using a previous genome assembly outweighs using a newer one. As suggested by the reviewer, we included the following statement in our revised manuscript:

To allow the required and meaningful comparability to previous whole genome sequencing studies in lymphomas (34,67-69), the human reference genome version GRCh37/hg19 was used.

The following publications were cited:

- 34 Hubschmann, D. et al. Mutational mechanisms shaping the coding and noncoding genome of germinal center derived B-cell lymphomas. *Leukemia*, doi:10.1038/s41375-021-01251-z (2021)
- 67 Arthur, S. E. et al. Genome-wide discovery of somatic regulatory variants in diffuse large B-cell lymphoma. *Nat Commun* 9, 4001, doi:10.1038/s41467-018-06354-3 (2018)
- 68 Morin, R. D. et al. Mutational and structural analysis of diffuse large B-cell lymphoma using whole-genome sequencing. *Blood* 122, 1256-1265, doi:10.1182/blood-2013-02-483727 (2013)
- 69 Zhang, J. et al. Genetic heterogeneity of diffuse large B-cell lymphoma. *Proceedings of the National Academy of Sciences of the United States of America* 110, 1398-1403, doi:10.1073/pnas.1205299110 (2013)

3.2 The equation for correcting telomere content will be difficult to understand to many readers. Please either cite the source from where this was obtained, or show its derivation, for example as can be readily done using an Expected Value approach for tumor and control and the assumption that EV(C) in the tumor is identical with EV(C) in the actual control sample.

Response: The telomere content was determined from whole-genome sequencing data using the software tool TelomereHunter (<https://doi.org/10.1186/s12859-019-2851-0>). The formula has been

previously applied to a Burkitt Lymphoma study by Lopez C. *et al.*, 2019 (PMID: 30926794). To clarify this, we now cite this publication and revised the text as follows:

Revised (in Material and Methods, Telomere content estimation): [...] *In the case of tumor samples, the telomere content was further corrected for the tumor cell content (TCC, as estimated by ATEseq) using the following formula as previously described (59), which corrects for inter-patient differences in telomere content assuming that the non-malignant cells in the tumor sample have the similar telomere content as in the control sample:*

$$T_{TCCcorrected} = (T - C(1 - TCC)) / TCC$$

The following publication was cited:

- 59 Lopez, C. *et al.* Genomic and transcriptomic changes complement each other in the pathogenesis of sporadic Burkitt lymphoma. *Nat Commun* 10, 1459, doi:10.1038/s41467-019-08578-3 (2019)

3.3 Line 303: "single base substitution" is mis-spelled

Response: This has been corrected.

3.4 The authors seem comfortable with accuracy for their mutation calls having VAF > 10%, but readers will wonder about lower-VAF mutations. This needs to be thoroughly and convincingly addressed.

Response: We agree that quality assurance of the mutation calls is of central importance. Out of context, the low VAF filter may seem alarming, however, it is important to note that VAF had not been our only criteria to filter mutations. The SNV calling workflow is the same workflow as has been used in ICGC-PCAWG, and numerous other recent lymphoma studies (e.g. <https://doi.org/10.1038/s41375-021-01251-z>, <https://doi.org/10.1038/s41467-019-08578-3>). In the landmark Nature paper for ICGC-PCAWG (<https://doi.org/10.1038/s41586-020-1969-6>) it was shown that the DKFZ SNV and indel calling workflows had a higher overall precision than the BROAD Institute's Mutect2 workflow (**Point-by-point response Figure 5**). For mutations with a low VAF between 0.1-0.2 the precision of DKFZ workflows was >0.9 for SNVs, and >0.75 for indels. We hope that this alleviates the concerns of the reviewer.

[REDACTED]

Point-by-point response Figure 5. The precision/sensitivity of DKFZ SNV and indel calling workflows reported in the ICGC PCAWG study. Figure taken from <https://doi.org/10.1038/s41586-020-1969-6>.

3.5 There are quite a few heuristics used in the bioinformatics analysis, for instance lines 258-261. There should be discussions about how some of these were determined and how sensitive the results are to those choices.

Response: While indeed interesting and relevant, we believe that this falls outside of the scope of this study. Our rationale in workflow choice has been to maintain comparability of our results to those of large bodies of work such ICGC MMML-seq, ICGC-PEDBRAIN, ICGC-PROSTATE, which use the same data processing workflows. It is also interesting to note that the calls derived from our SNV calling workflow were also used to generate the COSMIC mutational signatures, again giving testament to the usability of results generated by them. While we hope that this justifies that the workflows do indeed produce meaningful and reliable results. A separate manuscript describing the DKFZ SNV and indel calling workflows is underway.

REVIEWERS' COMMENTS

Reviewer #1 (Remarks to the Author):

The authors have thoroughly addressed the comments/concerns of the primary review and have markedly improved the clarity of this version. I have no further comments.

Reviewer #2 (Remarks to the Author):

In this revision, the authors have performed additional analyses and clarifications in response to my prior comments particularly addressing areas of ambiguity leading to an improvement in the flow and emphasis of their observations.

Interesting that MYC is frequently mutated despite absence of translocations in PCNSL and indeed maybe linked to MYC protein expression, although this event occurs more frequently than mutations – the authors should include the frequency (17%), rather than state 'high frequency' in the revised text.

Overall, the work presented in this revised manuscript adds a relevant resource to community given the rarity of this lymphoma subtype.

Reviewer #3 (Remarks to the Author):

The authors have addressed major comments.

Point-by-point responses to the reviewers' comments

We have used different font colours to differentiate between the reviewers' comments (**black font colour**), our responses (**blue font colour**), and revised manuscript passages (**dark orange colour**). We uploaded a manuscript version, where all changes are highlighted in yellow.

Reviewer #1 (Remarks to the Author)

Expert in primary central nervous system lymphomas

1. The authors have thoroughly addressed the comments/concerns of the primary review and have markedly improved the clarity of this version. I have no further comments.

Response: We thank reviewer #1 for his/her positive evaluation of our revised manuscript.

Reviewer #2 (Remarks to the Author)

Expert in lymphoma genomics

In this revision, the authors have performed additional analyses and clarifications in response to my prior comments particularly addressing areas of ambiguity leading to an improvement in the flow and emphasis of their observations.

1. Interesting that MYC is frequently mutated despite absence of translocations in PCNSL and indeed maybe linked to MYC protein expression, although this event occurs more frequently than mutations – the authors should include the frequency (17%), rather than state ‘high frequency’ in the revised text.

Overall, the work presented in this revised manuscript adds a relevant resource to community given the rarity of this lymphoma subtype.

Response: We thank reviewer #2 for his/her additional valuable comment concerning the revised version of our manuscript.

As suggested, we revised the manuscript accordingly (Results, Driver mutation on PCNSL):

“A remarkable finding was the identification of MYC mutations in 17% of PCNSL in the absence of MYC translocations.”

Reviewer #3 (Remarks to the Author)

Expert in cancer genomics and bioinformatics

1. The authors have addressed major comments.

Response: We thank reviewer #3 for his/her positive evaluation of our revised manuscript.